

# Southwestward propagating quasi-biweekly oscillations over the South-West Indian Ocean during boreal winter

Sambrita Ghatak[1,*] and Jai Sukhatme[1,2,*]

[1]Centre for Atmospheric and Oceanic Sciences, Indian Institute of Science, Bangalore, India.
[2]Divecha Centre for Climate Change, Indian Institute of Science, Bangalore, India.
[*]These authors contributed equally to this work.

**Correspondence:** Sambrita Ghatak (pulu.dec@gmail.com)

**Abstract.** An analysis of outgoing longwave radiation (OLR) over the South-West Indian Ocean (SWIO) yields regular, poleward propagating, large-scale, convectively coupled systems of alternating cyclonic and anticyclonic circulation with a quasi-biweekly period during boreal winter. Composites from 10 years (2000/01 – 2009/10) of OLR and reanalysis data show well-formed rotational gyres that can be tracked from near the equator to almost 35°S appearing slightly west of Sumatra and going towards Madagascar, i.e., with mean southwest propagation. The gyres show a marked northwest-southeast tilt, giving rise to a northeast-southwest oriented wavetrain. The scale of the gyres is about 30°–35°, their period is 18–20 days and they have a westward phase speed of approximately $4~\mathrm{ms}^{-1}$. The group velocity of these wave packets is near-zero. Velocity fields with OLR indicate that maxima of moist convective activity lie in the northeast sector of the gyres (in the Southern Hemisphere), likely a result of both convergence and the poleward rotational advection of moist air. Wavetrains comprising the quasi biweekly oscillation (QBWO) are born near the equator with a barotropic profile; a first baroclinic form emerges as they move southward and couple with moisture. In their decaying stage, convective activity decreases and the systems regain an equivalent barotropic structure. A vorticity budget reveals that the $\beta$ effect plays a leading role in the propagation of the QBWO, though moist coupling (via stretching) is important in reducing the speed of propagation of this mode. Further, these two terms with horizontal advection account for much of the observed vorticity tendency. Finally, apart from their contribution to intraseasonal variability, moist convection and strong circulation anomalies in the QBWO lows (cyclonic gyres with negative OLR anomalies) — especially in combination with the vorticity of the background flow during the boreal winter season — are shown to provide favourable conditions for the genesis of tropical cyclones (TCs). In particular, depressions are spawned within QBWO lows, some of which mature into TCs that affect Madagascar, other SWIO islands and the coast of southeast Africa.

## 1 Introduction

The quasi-biweekly oscillation (QBWO), with a timescale between that of synoptic disturbances and the Madden-Julian Oscillation (MJO; Madden and Julian, 1971, 1972), is an important component of tropical intraseasonal variability (Kikuchi and Wang, 2009). Early reports of the QBWO were in the context of the Indian summer monsoon, with fluctuations of a 15–20 day





period noted in lower tropospheric meridional wind anomalies (Keshavamurty, 1972). From then on, a great deal of research has investigated the features, mechanisms, and influence of the QBWO over the Indian Monsoon region (IMR), the western
North Pacific (WNP), the South China Sea (SCS), and Indo-China (Krishnamurti and Bhalme, 1976; Krishnamurti and Ardanuy, 1980; Chen and Chen, 1993; Fukutomi and Yasunari, 1999; Chan et al., 2002; Chatterjee and Goswami, 2004; Mao and Chan, 2005; Kikuchi and Wang, 2009; Chen and Sui, 2010; Jia and Yang, 2013; Wang and Chen, 2017). In fact, in their review paper, Kikuchi and Wang (2009) have documented westward propagating QBWO activity of tropical origin over several geographic locations around the tropics.

From a dynamical perspective, in most of the studies, the QBWO of tropical origin has been interpreted as the gravest meridional mode equatorial Rossby (ER) wave, modified by mean flow and convective coupling (Chatterjee and Goswami, 2004; Kikuchi and Wang, 2009; Chen and Sui, 2010). Using idealized models, it was concluded that the unstable ER wave is mainly driven by convective feedback via frictional boundary level convergence, and the mean flow helps to explain the realistic structure of the wave (Chatterjee and Goswami, 2004). Given the marked westward movement of ER waves, the pronounced
poleward propagation of the QBWO in many regions (for example, WNP and SCS) is not particularly well understood. Utilizing reanalysis data it was suggested that poleward propagation of QBWO may be understood as the modulation of an ER wave by the mean monsoon circulation (Kikuchi and Wang, 2009; Chen and Sui, 2010). But, fundamentally, as this mode is convectively coupled, its interaction with moisture is of particular interest. Overall, interactive moisture is very important for understanding the structure and propagation of convectively coupled equatorial disturbances. For example, for the 30–60 day
period MJO, it is thought to form an essential piece of the dynamics that has been highlighted in recent times (Fu et al., 2006; Benedict and Randall, 2007; Maloney, 2009; Kiranmayi and Maloney, 2011; Adames and Kim, 2016). Specifically, the "moisture mode" paradigm has shown significant promise in uncovering the fundamental dynamics and the eastward propagation of the MJO (Raymond and Fuchs, 2009; Adames and Kim, 2016; Jiang et al., 2018; Zhang et al., 2020). Indeed, westward propagating tropical modes have been seen to emerge from tropical imbalances in moist shallow water initial value experiments
(Suhas and Sukhatme, 2020). In this context, though some studies have looked at the effects of coupling of moisture with ER waves (Sukhatme, 2014) and the QBWO (Tao et al., 2009; Li et al., 2020, 2021), details of these interactions are still being investigated.

Most analyses regarding QBWO have been carried out in locations where the system affects considerable landmass, like IMR or SCS. Physically, in those regions, land-atmosphere interactions (mechanical or thermal) can play a significant role and possibly
complicate the growth and propagation of this mode. In fact, a few studies have emphasized these very points (for example, Wang et al., 2017). Naturally, in the Indian Ocean region, most of the QBWO related work has focused on this mode during boreal summer, in the context of Indian summer monsoon (Krishnamurti and Ardanuy, 1980; Chen and Chen, 1993; Chatterjee and Goswami, 2004; Roman-Stork et al., 2019). Though, a few studies have noted wind fluctuations on the quasi-biweekly time scale and their influence on the central and South-West Indian Ocean (SWIO) during boreal winter (Sengupta et al., 2004;
Han et al., 2007). Indeed, the existence of convective disturbances on a quasi-biweekly timescale in this basin has been noted in some form or other (Jury and Pathack, 1991; Jury et al., 1991; Kikuchi and Wang, 2009; Fukutomi and Yasunari, 2013, 2014),





but studies that focus on the QBWO in this basin are very limited, and its movement, structural and dynamical aspects still remain elusive. In fact, in general the literature on zonal movement with possible poleward propagation of intraseasonal modes in the Southern Hemisphere tropics is itself quite sparse. Physically, the SWIO, with its lack of significant landmass provides
a natural test bed that removes the complications due to land interactions. In other words, coupling between moisture and equatorial modes might be more clear cut and that may help us to understand the basic features of this mode in a simpler manner. Further, these basic ingredients can guide the formulation of idealized models that aim to capture the essential dynamics of the QBWO.

Another, more practical significance of this mode is related to its possible influence on tropical cyclones (TCs) in the SWIO
during the boreal winter. On average, the number of TCs (by TC, we mean 'Tropical Storm' and 'Tropical Cyclone' both categories of SWIO basin classification) formed each year in this basin ranges from 9–10 to 12–13 depending on the period considered (Mavume et al., 2009; Muthige et al., 2018; Leroux et al., 2018). A majority of these storms develop during the boreal winter season, and affect Madagascar, other SWIO islands, and sometimes south-east African nations, causing widespread devastation (Vitart et al., 2003; Reason and Keibel, 2004; Fitchett and Grab, 2014; Leroux et al., 2018; Bousquet
et al., 2021). Most recently, cyclone Idai, which made landfall on the coast of Mozambique on March 2019, was one of the deadliest TCs on record and brought humanitarian crisis in parts of south-east Africa, namely Mozambique, Zimbabwe and Malawi (Warren, 2019; Barthe et al., 2021). Currently, the track and strengths of TCs in SWIO are difficult to predict accurately more than a few days ahead (Kolstad, 2021), making it difficult to provide the time required for evacuation and other preparedness measures (Webster, 2013; White et al., 2017; Mavhura, 2020). But, in the SWIO, TC modulation by convectively
coupled ER waves with quasi-biweekly time-period is statistically significant (Bessafi and Wheeler, 2006), and given the 8–12 day predictability horizon for these waves (Li and Stechmann, 2020), this signal can be exploited for advance prediction of TCs. Thus, a better understanding of the QBWO, possibly in terms of convectively coupled ER waves, and their connection with TCs in this region, is relevant for improved forecasts of cyclones in the SWIO basin. In fact, the influence of QBWO on tropical cyclogenesis is well documented in other ocean basins (Ling et al., 2016; Zhao et al., 2016). Further, apart from
influencing extreme events such as TCs, this mode also has a bearing on the regional rainfall pattern. Specifically, a 10–20 day cycle in boreal winter rainfall is documented in Madagascar (Nassor and Jury, 1998). Westward propagating transient waves of similar timescale coming from SWIO have also been observed to contribute to fluctuations in rainfall in the south-east African coastal region, and to cause heavy rainfall episodes even in the interior of the continent, such as the Southern African Plateau (Jury and Pathack, 1991; Jury et al., 1991). It is quite possible that QBWO has a role to play in these observed rainfall cycles.
In turn, this provides a valuable framework for active and break cycle prediction in Madagascar and south-east Africa that predominantly depends on rain-fed agriculture (Malherbe et al., 2012; Silva and Matyas, 2014; Macron et al., 2016; Pohl et al., 2017; Rapolaki et al., 2019). In addition to this, a better understanding of QBWO in this basin is possibly connected to the MJO prediction; specifically, much like the influence of ER waves on the genesis of the MJO (Roundy and Frank, 2004b), the QBWO too could influence intraseasonal activity on longer timescales in the SWIO.





The most detailed work on sub-monthly intraseasonal variability till date over this broad geographical region in boreal winter was done by Fukutomi and Yasunari (2013, 2014). These authors used Japanese reanalysis (JRA25-JCDAS) data to examine variability in a broad 6–30 day window. They found wavetrains oriented in a northeast-southwest direction from Sumatra towards Madagascar, with a wavelength of 3000–5000 km, and westward-southwestward phase propagation. These were interpreted as combinations of ER and mixed Rossby-gravity (MRG) gyres, propagating in a favorable background of monsoon westerlies. But, effects of coupling with moisture have not been investigated, nor was a time period range of quasi-biweekly activity (without smaller timescale synoptic influence) considered in isolation. Moreover, the propagation and dynamical ingredients required for the formation of the QBWO in this region have not been clearly identified, nor has the influence of this mode on TCs been elucidated.

In this paper, we aim to fill some of these gaps; after a description of the data and methods used in the study (Section 2), we proceed with a detailed documentation of the characteristic features of the QBWO in the SWIO (Section 3), including its spatial and temporal scales, propagation, horizontal and vertical structure and evolution. Our analysis is based on composites generated from ten years of daily reanalysis data. With the characterisation in hand, we then proceed to a vorticity balance (Section 4) to examine the features involved in the maintenance and propagation of this mode. We then consider the influence of the QBWO on tropical depressions and TCs (Section 5); in particular, a detailed case study of the 2008–09 winter season is presented to show the evolution of the QBWO wavetrain and the concomitant birth and development of depressions, which eventually mature into TCs in the convective environment of the quasi-biweekly system. Finally, in Section 6, a summary and discussion of our results concludes the paper.

## 2  Data and Methodology

Daily mean products from the National Centers for Environmental Prediction/National Center for Atmospheric Research (NCEP/NCAR) reanalysis project serve as the main data set for this study (Kalnay et al., 1996). Specifically, we have used 10 years of horizontal winds and vertical velocity data at 12 pressure levels (1000 to 100 hPa) that spans December to March (DJFM) from 2000/2001–2009/2010 and has a horizontal resolution of 2.5°. This data is used to calculate the derived fields presented in this paper. Some fields, such as relative vorticity and various terms of the vorticity budget are computed using Windspharm package (Dawson, 2016). Daily outgoing longwave radiation (OLR) data from the National Oceanic and Atmospheric Administration (NOAA) satellites serves as a proxy for moist tropical convection (Liebmann and Smith, 1996). 4-times daily precipitable water (PW) data is used from ERA-Interim reanalysis (Dee et al., 2011), from which daily average is calculated. Both, PW and OLR have a spatial resolution of 2.5°. In addition, wind (at 850 hPa) and OLR data at 0000UTC of specific days are taken from hourly wind data of ERA5 (Hersbach et al., 2020), and 8-times daily OLR of the Kalpana satellite (Mahakur et al., 2013) respectively, to examine specific events associated with TCs, these data are at a much finer spatial resolution of 0.25°. Precipitation data is from the Tropical Rainfall Measuring Mission (TRMM), specifically the product 3B42 is used at a daily resolution (Huffman et al., 2007). This data also has a spatial resolution of 0.25°. Finally, to examine



the relation between the QBWO and cyclones, we have used tropical cyclone trajectory data from the Joint Typhoon Warning Centre (JTWC) best track repository, which is available at 4-times daily frequency.

To isolate the QBWO signal, we used a filter with a 14–30 day band. Most studies regarding QBWO use a 10–20 day filter, but
there are variations, such as 12–20 day (Kikuchi and Wang, 2009; Jia and Yang, 2013), 10–25 day (Fukutomi and Yasunari, 1999), 12–24 day (Chen and Chen, 1995) and 6–30 day (Fukutomi and Yasunari, 2013, 2014) in particular studies. Here, we use a lower limit of 14 day to exclude synoptic scale signals, and the upper limit of 30 day is used instead of widely used 20 day as previous studies have shown that there can be systems of QBWO scale which have a slightly larger period than 20 days (Molinari et al., 2007). It turns out that the dominant period around the two-week scale is actually near 20 day in the SWIO,
hence, for the upper threshold, the use of 30 day (instead of the widely used 20 day cutoff) seems appropriate, as it doesn't exclude signals which might have periodicity slightly larger than 20 days. We note that significant power associated with MJO is well above 30 day, so we don't run into the risk of including MJO signals while using a 14–30 day band window.

To extract 14–30 day variability, a Lanczos band-pass filtering method is used (Duchon, 1979). Further, to highlight the QBWO, the annual cycle of the time series in question is removed by subtracting the mean and the first three Fourier harmonics before
filtering is done. To check robustness, we varied the filtering band and it produces essentially identical results up to a 10–35 day range, so our result is not sensitive to the width of the filter as long as it satisfactorily isolates QBWO signal from the variability of other tropical modes. To describe the structure and evolution of the QBWO, we have constructed composites of OLR and low-level (850 hPa) circulation from the filtered data. Members for the composites satisfy the following three criteria: 1) In the chosen box (52.5°–60° E, 10°–17.5° S; Figure 1) — a region of high 14–30 day variability (in terms of standard deviation)
of OLR anomaly — there should be complete and distinct phase cycle of averaged 14–30 day OLR anomaly, i.e., a distinct maximum should monotonically decrease to a distinct minimum and then again monotonically increase to a distinct maximum. 2) The negative anomaly of box-averaged OLR on Day 0 (which is the day when the box-averaged OLR is locally minimum) must be less than −1 standard deviation; and similarly the previous and subsequent maxima of positive anomalies of box-averaged OLR must be more than 1 standard deviation. 3) The identified minima should be outside the first and last 16 days
of the season. This condition is used as we leave 16 days before and after the minima based on the filter to ensure that there is room for a complete cycle within the season (DJFM). These criteria help us to identify strong convectively coupled systems and result in 19 members from 2000/01–2009/10 in the composite. We also varied the base region a bit to check whether the results depend strongly on the particular box chosen. Indeed, as long as the base box remains inside the region of high filtered OLR standard deviation, the composite structure is not affected. Further, the composite vertical structure and vorticity budget
of these systems are also analyzed to understand the evolution of the QBWO and its propagation. A detailed description of these analysis methods are given in the respective sections.



## 3 Boreal Winter Quasi-biweekly Activity

The winter season 14–30 day variability of organized moist convection in terms of the standard deviation of filtered OLR is shown in Figure 1, along with mean background PW. Quite clearly, in the SWIO, the quasi-biweekly part of intraseasonal

activity is contained within regions of high PW (Sukhatme, 2013). Further, the north-east to south-west(NE-SW) tilt of moist convective activity in the SWIO appears to follow that of PW and suggests that this could be a pathway for the propagation of quasi-biweekly intraseasonal systems in this region.

This possible southward propagation is exemplified in Hovmoller diagrams of OLR and rainfall anomalies presented in Figure 2. The winters chosen are 2001–02 (first panel from above), 2006–07 (second panel from above) and 2008–09 (third panel

from above), though this type of movement is seen in almost every year. Examining the OLR anomalies (contours in Figure 2), propagation begins near the equator and extends up to about 35°S, with some degree of variability. Further, wavepackets in the form of alternating highs and lows in the filtered OLR anomaly are visible in each of these cases — specifically, from Day 70 to 115 in 2001–02, Day 80 to 120 in 2006–07 and Day 40-120 in 2008–09. From these diagrams, one can note that these systems are slightly more probable in the second half of the season, i.e., late winter. This is also by and large true of

other winters in the 10-year data we have examined. The phase speed of these disturbances is negative indicating southward movement but some wavepackets suggest a small positive or northward group velocity (for example, in 2008–09 from Day 60 to 90). We will return to quantification of these issues when we construct composites of the southward moving systems. It should be kept in mind that we have shown specific years in Figure 2, but in fact, cohesive southward propagating events are observed in almost every winter season. The colors in Figure 2 show rainfall anomalies, and negative OLR contours contain

positive (blue) precipitation anomalies, or enhanced rainfall. In most cases, precipitation anomalies are large near the equator, though in some cases (for example, 2001-002, Days 100–110) large deviations of rainfall from the mean state are observed as far as 35°S, i.e., well into the subtropical SWIO. Note that, even during the winter, we observe some instances of northward propagation (for example, between Days 15 and 40 in 2006–07), albeit weaker, in this intraseasonal window (Srinivasan and Smith, 1996). Though the movement of such systems is restricted from the equator to about 10°N.

## 175 3.1 Horizontal Structure

Composites of the southward moving intraseasonal systems, constructed using the techniques outlined in the Data and Methodology section, are presented in Figure 3. On Day −9, northeast of Madagascar, we see a well-developed anomalous anticyclonic circulation coupled with positive OLR anomalies (red color in Figure 3) indicating a lack of moist convection. Meanwhile, near the equator, we see an emerging low with anomalous westerlies between 50°–90°E and scattered negative OLR anomalies.

North of the equator too, we see a hint of a cyclonic circulation, with anomalous westerlies hugging the equator between similar longitudes, and negative OLR anomalies (though weaker in magnitude than the Southern Hemisphere counterpart). By Day −6, the newly formed cyclonic vortex near the equator becomes more prominent, gets a little tilted and a small area of strong anomalous convection is seen in its northeast sector. North of the equator, the weak cyclonic anomaly has almost vanished but

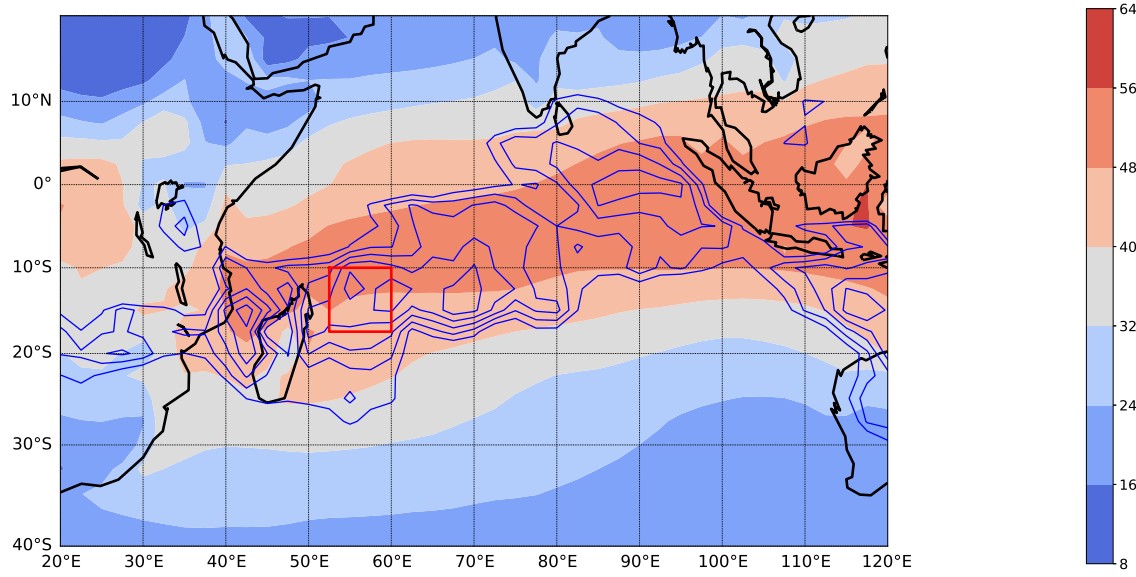

**Figure 1.** Geographical distribution of mean precipitable water (kg m$^{-2}$) in shading and the standard deviation of 14–30 day filtered OLR (W m$^{-2}$) in contours for the boreal winter (DJFM) during the period 2000/2001–2009/10. The red box is the region chosen for composite analysis. The contour interval for the standard deviation of OLR is 1W m$^{-2}$, starting from 15W m$^{-2}$.

weak negative OLR anomalies are still visible. Simultaneously, near Madagascar, the anticyclonic circulation and associated

positive OLR anomalies have moved to the southwest, expanded but become weaker and almost engulfed the entire island and made some inroads into south-east Africa, especially parts of the coast and the Great Rift Valley. In effect, a clear NE-SW oriented wavetrain pattern has been established.

Moving ahead, by Day −3, the cyclonic anomaly expands with a well-defined vortex, moves southwest and reaches the northern tip of Madagascar. Further, this circulation is now coupled with strong anomalous convection within the vortex. This southwest

movement continues and by Day 0, the cyclonic gyre covers most of Madagascar, is tightly coupled with moisture resulting in intensified moist convection and large negative OLR anomalies. In succession, near the equator, we now see the hint of the birth of the next high accompanied with anomalous easterlies and suppressed convection (positive OLR anomalies) between 50°E to 90°E. By Day 3, the convective cyclonic gyre moves further southward, expands meridionally and starts losing its well-defined shape. In fact, convection associated with the cyclonic lobe extends into south-east Africa , particularly the regions mentioned

above. A clear area of suppressed convection with anticyclonic flow starts forming near the equator to the north-east of the cyclonic low, having strong positive OLR anomalies to the northeast of the anticyclonic lobe just south of the equator between 50°E to 80°E. Again, a NE-SW oriented wavetrain is clearly visible. Thus, we can say, from Day −6 to Day 3, the oscillation



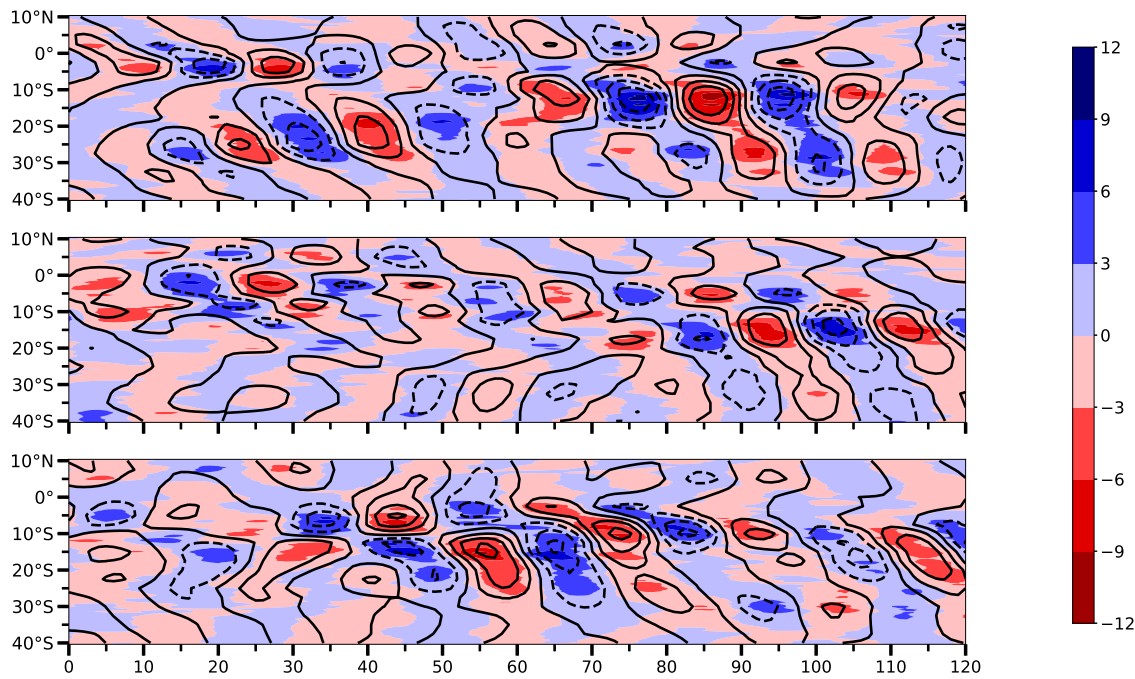

**Figure 2.** Time-latitude diagram of 14–30 day filtered OLR (W m$^{-2}$; contours) and precipitation anomalies (mm/day; shading) for 3 individual years. 2001–02 (upper panel), 2006–07 (middle panel) and 2008–2009 (lower panel), averaged over $40°$–$70°$ E. The contour interval for OLR anomalies is 10W m$^{-2}$, contours signifying positive and zero values are solid, those for negative values are dotted. The time axis goes from December to March (121 days).

approximately completes a half cycle. On Day 6, the cyclonic gyre moves further south and crosses $30°$S, weakens and loses its convective signature. The area of suppressed convection associated with the anticyclonic vortex to the north intensifies and

follows the evolution of the preceding low, of course with the opposite sign. The thick black line in Figure 3 (Day $-6$) shows the approximate track of the QBWO, and is constructed by connecting vorticity minima of the cyclonic gyres on Days $-6, 0$ and 6.

Thus, the low level (850 hPa) circulation anomalies exhibit southwest-southward propagation of QBWO over the SWIO. The system starts near the equator between roughly $50°$–$90°$E, moves southwestward and then exhibits a jump towards the south.

While moving away from the equator, the QBWO matures and gets tightly coupled with moist convection. In fact, at this stage, we note a clear northwest-southeast(NW-SE) tilt associated with its gyres, and alternating cyclonic and anticyclonic circulation patterns together give rise to a NE-SW oriented wavetrain. Finally, the oscillation dies down southeast of Madagascar, near approximately $30°$–$35°$S traversing a total meridional distance of about 30 degrees. Further, the area of negative or positive OLR anomalies, as well as the cyclonic or anticyclonic circulations, cover around 30–35 degrees of longitude while the lati-

tudinal extent is approximately 20 degrees. This suggests that the zonal wavelength of the system is roughly 6500–7500 km





(or, wavenumber 5–6). These lower tropospheric horizontal features that we have catalogued here are similar to that of boreal summer QBWO (Chen and Sui, 2010; Wang and Chen, 2017), and apart from a larger spatial scale, they also have a striking resemblance with $n = 1$ ER waves (Molinari et al., 2007). In fact, the QBWO over WNP has been interpreted as a modified ER wave (Chen and Sui, 2010). Our composite structure is also quite similar to that of the observed ER waves (Wheeler et al.,

2000), except for the striking asymmetry about the equator and somewhat larger poleward movement. A distinguishing feature of the QBWO over SWIO is that the strongest convection — or, most negative OLR anomaly — is located in the northeast sector of the cyclonic circulation. This negative OLR anomaly is likely due to both convergence between the gyres as well as horizontal rotational poleward advection of moist air out of the equatorial region. This too is reminiscent of observations of ER waves (Wheeler et al., 2000; Molinari et al., 2007) and has been noted in recent idealized moist shallow water simulations

(Suhas and Sukhatme, 2020).

To estimate the period and speed of propagation of the QBWO, we now construct time-latitude and time-longitude Hovmöller diagrams. Specifically, two panels in Figure 4 show the time-latitude diagrams of filtered OLR and zonal wind anomalies averaged over $40°$–$70°$ E, both depicting a southward phase speed of approximately $1.4°$/day. Similarly, the two panels in Figure 5 show the longitude-time diagrams of filtered OLR and meridional wind anomalies averaged over $5°$–$25°$ S. The OLR

anomalies show that from Day $-8$ to $4$ or so, there is a westward speed of approximately $3°$/day, but after that, this movement ceases. This is in accord with the composite which suggests that from Day 3 to Day 6, southward propagation dominates over the westward motion. Consistent with this, the meridional wind anomalies averaged over the same area also show a westward speed of about $3°$/day or $4$ m s$^{-1}$. The group velocity that can be estimated from 4 and 5 is quite clearly close to zero. The wavelength that can be estimated here also matches with our estimation discussed above. The time period from both these

figures is also consistent, i.e., close to 18–20 days. Closer to the equator, Figure 6 shows the longitude-time diagram of filtered zonal wind anomalies averaged over $0°$–$10°$S. Here, the movement captured is slightly faster, in fact, westward propagation of approximately $3.3°$/day is evident. The wavelength also appears to be slightly larger than what can be seen from Figure 5. The apparent disagreement between the westward phase speeds and wavelengths from zonal and meridional winds of the same system seems paradoxical. But, in fact, this is quite similar to the properties of ER waves noted in different parts of

the tropics (Molinari et al., 2007), and the difference is attributed to the changes in the background flows in equatorial and off-equatorial regions. As seen in Figure 7, the low-level background winds have a cyclonic shear with westerly flow near the equator between $50°$E to almost $90°$E, and an easterly further poleward, creating a patch of strong cyclonic vorticity, with a monsoon trough-like character.

In all, our composites are quite similar to Northern Hemisphere counterparts such as those from the SCS or the WNP (Chen and

Sui, 2010; Wang and Chen, 2017), where QBWOs are asymmetric about the equator, i.e., mostly confined in one hemisphere (the summer hemisphere) and usually understood as modified ER waves. Indeed, ER waves observed by Molinari et al. (2007) also have a similar hemispheric asymmetry. As suggested in these studies, this shift from the theoretical symmetric structure (Matsuno, 1966), is likely due to the prevalent background flow. Further, in the boreal winter, PW is concentrated in the Southern Hemisphere which points to the possibility of moist coupling and growth south of the equator. The wavelength (around



6500–7500 km) and time-period (18–20 days) in our observations are comparable to the range of estimates for convectively coupled ER waves (Numaguti, 1995; Wheeler et al., 2000; Janicot et al., 2010). Kiladis and Wheeler (1995) also observed ER waves with similar wavelengths, but slightly smaller time-periods and faster phase speeds, though this might be an artefact of focusing on circulation features rather than strongly convectively coupled cases (Wheeler and Kiladis, 1999). These scales also match with observations of the westward propagating QBWO (Chatterjee and Goswami, 2004; Kikuchi and Wang, 2009; Chen and Sui, 2010). In fact, the observed wavelength and time-period lie well within the designated region for convectively coupled ER waves (Wheeler and Kiladis, 1999; Wheeler et al., 2000; Roundy and Frank, 2004a). Naturally, the phase speed is also comparable to that of observed westward propagation of convectively coupled ER waves (Numaguti, 1995; Wheeler and Kiladis, 1999; Wheeler et al., 2000; Bessafi and Wheeler, 2006; Kiladis et al., 2009; Janicot et al., 2010) as well as westward propagating QBWOs (Chatterjee and Goswami, 2004; Chen and Sui, 2010). Quite remarkably, the length and time scales, as well as phase velocity observed here match with that of westward propagating equatorial modes in an idealized moist shallow water system (Sukhatme, 2014). The movement of wave packets (Figure 2) suggests a near-zero group velocity which also agrees with observations of ER waves (Wheeler and Kiladis, 1999; Wheeler et al., 2000; Molinari et al., 2007) and the boreal summer QBWO (Chen and Sui, 2010; Wang and Chen, 2017). For an equivalent depth range of 12–50 m (Wheeler and Kiladis, 1999), the theoretical $n = 1$ ER wave with a typical wavelength of 6000 km and a westward phase speed of 4–5.5 m s$^{-1}$ and group velocity of 0.8–2.6 m s$^{-1}$, with no background flow. Here, the estimated phase velocity is in close agreement with theory, though the group velocity is somewhat smaller. Of course, the QBWO in SWIO is subjected to a complex background flow with horizontal and vertical shear, and we don't expect the theoretical phase and group velocity estimates to exactly match real-world observations. Thus, much like its boreal summer counterparts, the properties and horizontal structure of the boreal winter QBWO bear a close relation with ER waves in the SWIO region. Indeed, observed ER waves in the SWIO (for example, Figure 13 in Bessafi and Wheeler, 2006) bear a close resemblance to our composite.

Notably, these results are somewhat different than those presented by Fukutomi and Yasunari (2013) in their study of sub-monthly scale intraseasonal activity in the SWIO. They found meridionally elongated cyclonic and anti-cyclonic vortices and documented continuous northeast-southwest propagation, whereas we observe an initial southwest movement of zonally elongated gyres followed by predominantly southward propagation. Further, unlike our analysis, they found wavetrains to cross the equator. Though the estimated phase speed in their study is close to ours, they noted an eastward group velocity of 8 m s$^{-1}$, which we clearly do not observe in our composite. These differences may be an artefact of using different temporal windows, as their analysis likely included both mixed Rossby-gravity and ER signals.

## 3.2 Vertical Structure

With a description of horizontal composites, we now examine the vertical structure of QBWO using data on 12 pressure levels that range from 1000 to 100 hPa. Specifically, vertical cross-sections of relative vorticity and vertical wind associated with QBWO along the track line in Figure 3, from Day −9 to Day 6, are shown in Figure 8. A well-developed train of positive and negative vorticity anomalies can be seen on all days. Specifically, if we follow the negative vorticity anomaly in the lower





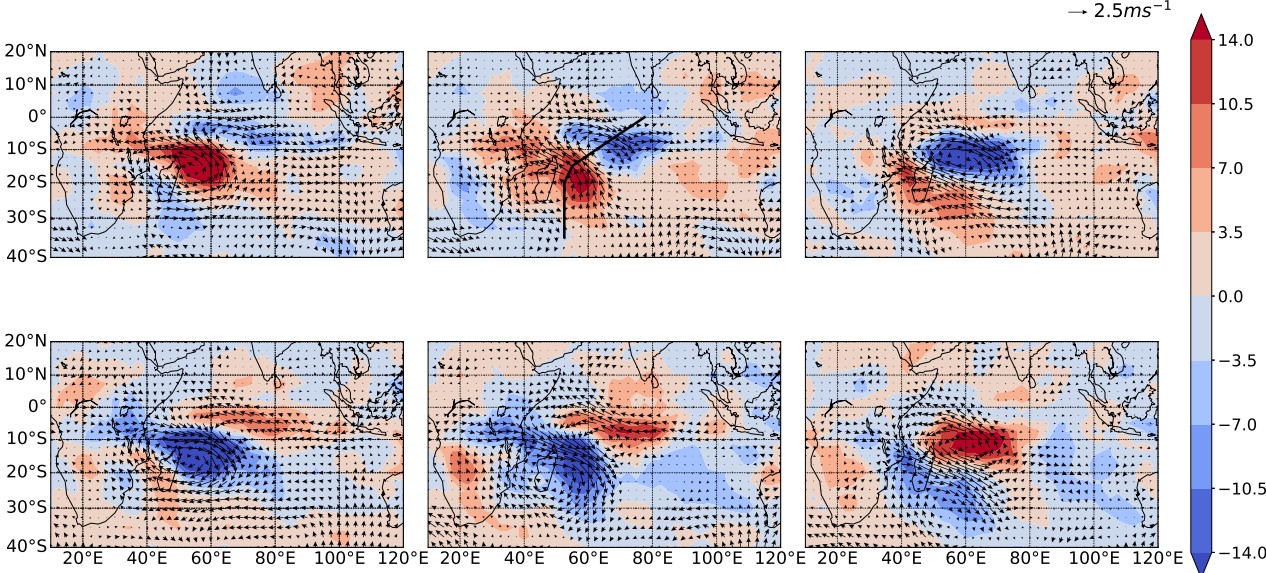

**Figure 3.** Composite of 14–30 day filtered OLR (W m$^{-2}$) and 850 hPa wind anomalies for the boreal winter(DJFM) from Day $-9$ to Day 6. Specifically, Days $-9, -6, -3$ are in the upper row (from left) and Days 0, 3 and 6 are shown in the lower row (from left). These are constructed from events during the period 2000/01–2009/10. The solid black line represents the approximate track of the composite of these events.

troposphere, we can observe its evolution in time and space. On Day $-9$, when the negative anomaly centre in the lower troposphere is near 10°S, the system has a barotropic structure, i.e., contours of the negative vorticity anomaly reach straight

280 up to about 100 hPa from the surface, though, the anomaly extends slightly towards the southwest in the upper troposphere near 200 hPa. One can note, at this stage, the negative vorticity lobe is not associated with convection. By Day $-6$, this anomaly has barely moved but it has intensified, particularly up to about 400 hPa,and the barotropic shape (no change of sign of perturbation up to 100 hPa) remains, though we can see a more prominent southwest extension of the negative anomaly at the upper levels from about 300 hPa. By Day $-3$, the negative anomaly at lower levels has moved southwestward and expanded, and the system

285 is no longer barotropic in structure. In fact, the anomaly extends up to 400 hPa and then changes the sign. This negative anomaly tilts towards the southwest from about 400 hPa and resembles a first baroclinic mode. At this stage, the cyclonic vortex at the lower level is associated with strong convection as can be deduced from the OLR anomalies on Day $-3$ in Figure 3.

By Day 0, the system moves further southwestward, the negative vorticity anomaly strengthens in the lower and the middle troposphere and maintains its tilt and baroclinic structure. Now by Day 3, the lower troposphere negative anomaly centre

290 continues its propagation and the area it covers expands up to 35°S, which is the southernmost point of the track in Figure 3. But, from Day 0 to Day 3, along with expansion, the anomaly weakens considerably and pushes slightly upwards, almost up to



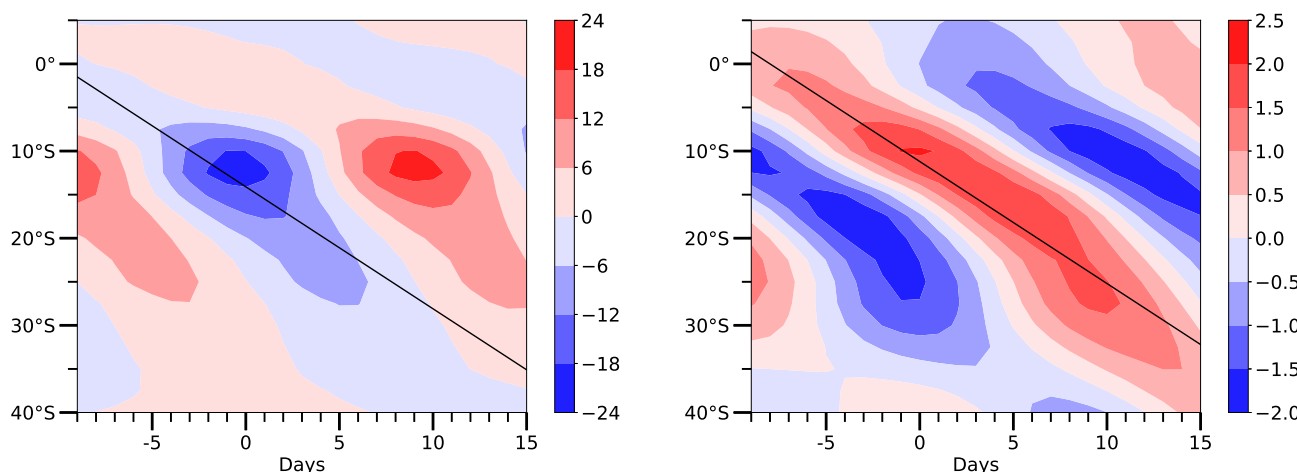

**Figure 4.** Time–latitude diagram of 14–30 day filtered OLR (W m$^{-2}$) and zonal wind (m s$^{-1}$) anomalies averaged over 40°–70°E from the composite in Figure 3. The black lines in both the panels correspond to a southward movement of 1.4 °/day.

200 hPa; so its overall profile has more of an equivalent-barotropic form. Finally, by Day 6, the centre weakens further, moves towards the edge of the track and can be seen to have regained a near barotropic profile near about 30°S.

The panels in Figure 8 are arranged such that the snapshots in the upper and lower rows are 9 days apart; clearly, we see
295 similar structures in the rows but of opposite signs, which is expected after approximately half-period of the QBWO. So, the positive vorticity anomaly (red in Figure 8) goes through a similar life cycle as described in the previous two paragraphs for the negative anomaly. To summarize, the QBWO starts with an almost barotropic structure near the equator when it is not strongly coupled with moist convection. In its mature stage, when coupled with convection, it strengthens, changes progressively from an equivalent barotropic to a first baroclinic profile and tilts towards the southwest. Towards the end, convection dies down,
300 the system becomes weak and comes back to a barotropic structure. These observations are in line with findings on ER waves in different parts of the global tropics (Wheeler et al., 2000; Yang et al., 2007; Kiladis et al., 2009), where it is seen that convectively uncoupled/weakly coupled waves have a near barotropic structure, and convectively coupled waves have a more baroclinic structure.

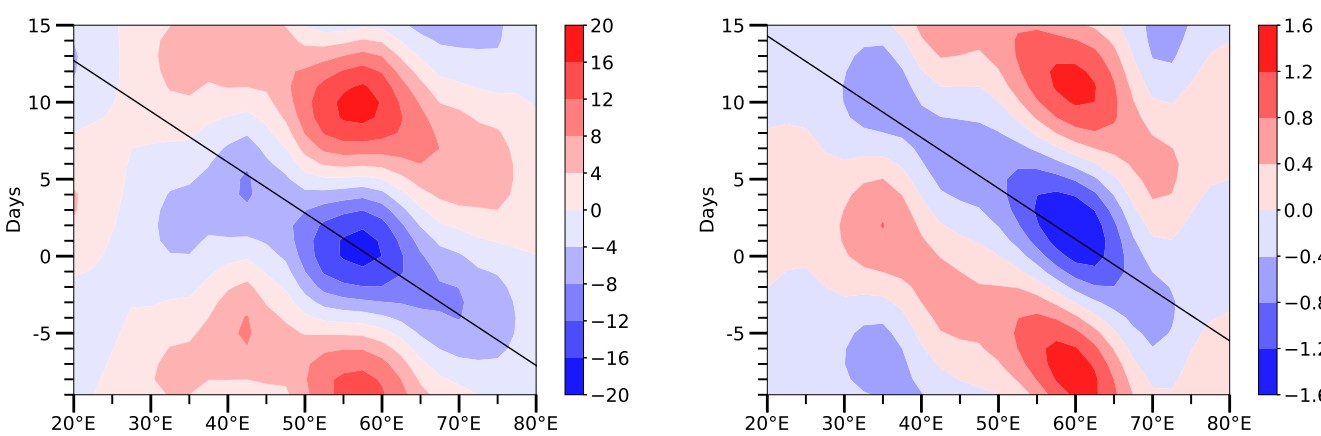

**Figure 5.** Longitude–time diagram of 14–30 day filtered OLR (W m$^{-2}$) and meridional wind (m s$^{-1}$) anomalies averaged over 5°–25°S from the composite in Figure 3. The black lines in both the panels correspond to a westward movement of 3°/day, respectively.

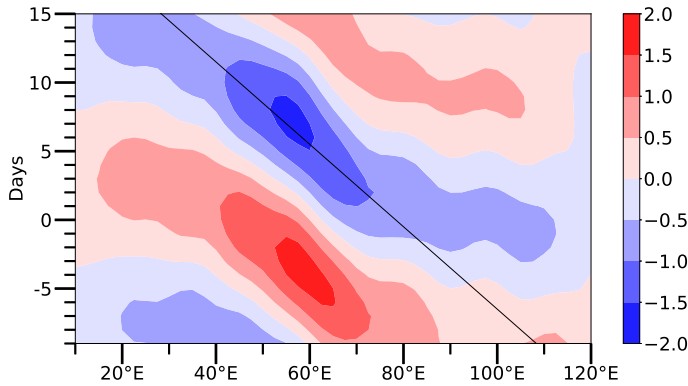

**Figure 6.** Longitude–time diagram of 14–30 day filtered zonal wind (m s$^{-1}$) anomalies averaged over 0°–10°S from the composite in Figure 3. The black line corresponds to a westward movement of 3.3°/day.

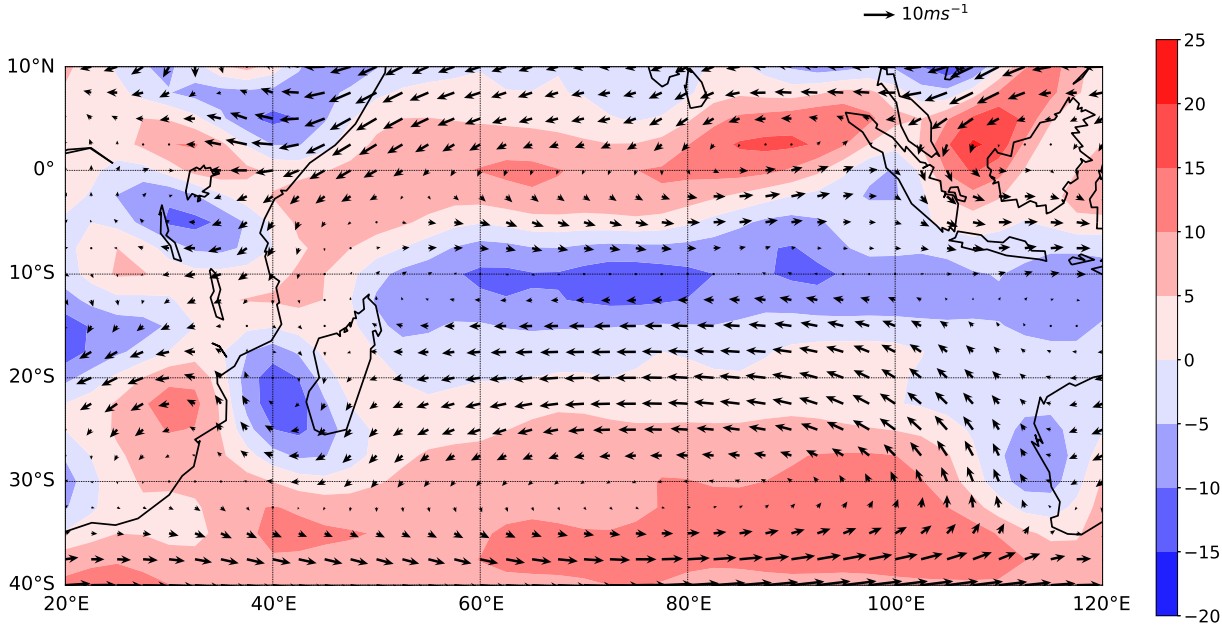

**Figure 7.** Mean background wind at 850 hPa and associated relative vorticity ($10^{-6}$s$^{-1}$; shading). This is the time-average for the boreal winter (DJFM) for all 10 years.

Interestingly, the configuration of vertical velocity and vorticity is somewhat different near and away from the equator. In particular, near the equator, upward and downward motion is located at the northeast corner of the negative and positive vorticity anomalies, respectively, i.e., displaced from the centre of the gyre. We can see this clearly on Day $-9$ and Day $0$ where the negative and positive anomaly centres are near 10°S, respectively. This is also in agreement with the positions of the strongest convection in Figure 3. When the system moves away from the equator, vertical motion is almost aligned with the vorticity anomalies. Much like the observation by Chen and Sui (2010) in the WNP, this behavior strengthens the association of the QBWO with equatorial modes such as ER and MRG waves that also show a phase lag between the circulation centre (vorticity anomalies) and convergence (vertical velocity). But, other features of MRG waves don't match those observed here in our study, thus pointing towards the intimate connection of the QBWO in the boreal winter with large-scale ER wave modes.



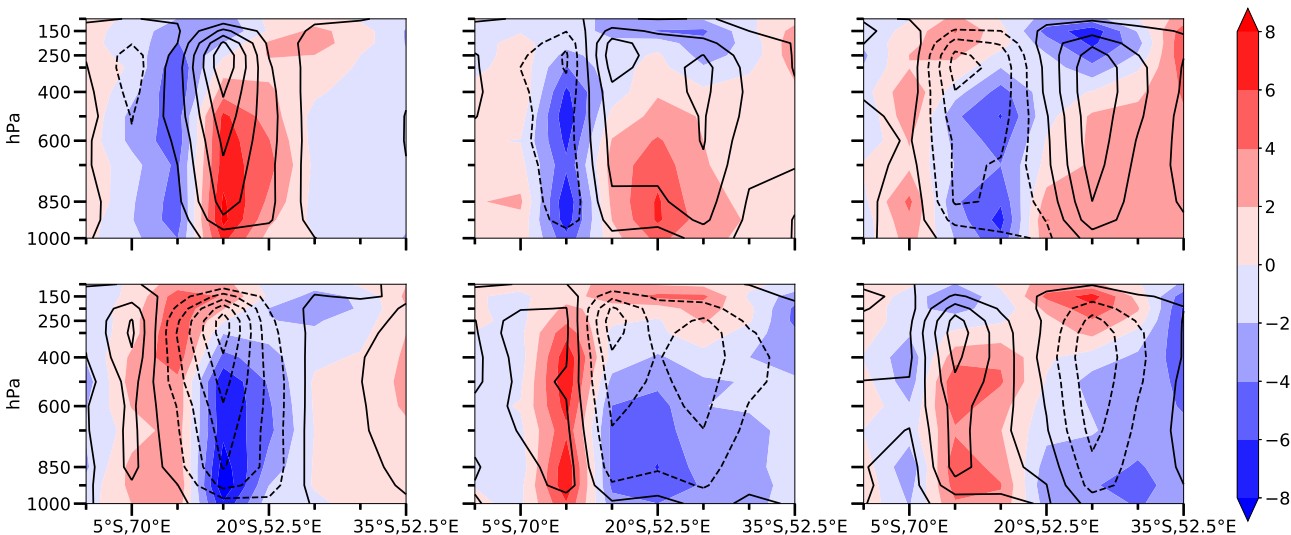

**Figure 8.** Vertical cross sections of the composite 14–30 day filtered vorticity anomalies ($10^{-6}$s$^{-1}$) in shading with vertical velocity anomalies (Pa s$^{-1}$) in contours following the approximate track-line from the composite in Figure 3. The contour interval for vertical velocity anomaly is $0.01$Pa s$^{-1}$, contours signifying positive(downwards) and zero values are solid, and those for negative (upwards) values are dotted. The upper panel shows Day $-9$, Day $-6$ and Day $-3$ (from left) while the lower panel shows Day 0, Day 3 and Day 6 (from left) fields.

## 4  Vorticity budget of the QBWO

We now analyse the propagation and dynamics of the QBWO by constructing a budget of the vorticity anomalies of the system.
The relevant equation reads (Wang and Chen, 2017),

$$\frac{\partial \zeta'}{\partial t} = \left(-\omega \frac{\partial \zeta}{\partial p}\right)' + (-\mathbf{V}.\nabla_h \zeta)' + \left(-v\frac{\partial f}{\partial y}\right)' + [-(\zeta + f)D]' + T' + R', \tag{1}$$

where, $\zeta = \left(\frac{\partial v}{\partial x} - \frac{\partial u}{\partial y}\right)$ and $D = \left(\frac{\partial u}{\partial x} + \frac{\partial v}{\partial y}\right)$ are the relative vorticity and divergence, respectively. $\mathbf{V} = u\mathbf{i} + v\mathbf{j}$ is the horizontal wind, $\nabla_h = \mathbf{i}(\frac{\partial}{\partial x}) + \mathbf{j}(\frac{\partial}{\partial y})$ is the horizontal gradient operator, $f$ is Coriolis parameter and $\omega$ is the vertical velocity in pressure co-ordinates. Here, prime denotes a 14–30 day anomaly. $T'$ is given by $[(\frac{\partial \omega}{\partial y})(\frac{\partial u}{\partial p}) - (\frac{\partial \omega}{\partial x})(\frac{\partial v}{\partial p})]'$, which is the anomaly of the tilting term, $[-(\zeta + f)D]'$ represents the anomaly of the stretching term, $\frac{\partial \zeta'}{\partial t}$ is the local tendency of the vorticity anomaly, $(-\mathbf{V}.\nabla_h \zeta)'$ and $(-\omega \frac{\partial \zeta}{\partial p})'$ represent anomalies of the horizontal and vertical advection of relative vorticity, respectively, and $(-v\frac{\partial f}{\partial y})'$ is the anomaly due to the $\beta$ effect. The last term, $R'$, is the residual, which includes contributions from all processes not expressed explicitly in Equation 1.





We compute composite means of individual terms in this budget, i.e., we first compute the terms for each day, and then take the mean over all the cases included in the composite. For Figure 9, we adopt an Eulerian view and spatially average the composite mean of various terms in Equation 1 over the blue box marked in Figure 1. These terms are then superimposed on the composite mean of box averaged vorticity anomalies. Whereas, in Figure 13, we follow the approximate track of the system as shown by the black line in Figure 3, and plot the time evolution of the 850–600 hPa averaged composite mean of various terms of the vorticity budget along with vorticity anomalies. Figures 10-12 depict three individual days (Days $-3$, $0$ and $3$) from the composite for detailed examination and show the composite mean of dominant terms in the lower tropospheric vorticity budget.

The budget in Figure 9 shows that in the positive QBWO convection phase (associated with above-average convection, near Day 0 in the figure), relative vorticity anomalies (contours) are negative below 300 hPa and positive above that level. The negative QBWO phase (associated with below average convection) shows the opposite pattern. At these lower levels, negative vorticity anomalies begin to develop on Day $-5$, attain a maximum by Day $-2$ and decay by approximately Day 3 which is in agreement with the lower level circulation described in Figure 3. After that, the cycle begins again and positive vorticity anomalies start to develop. The tilting and vertical advection terms (Figure 9 lower panel) are small (one order smaller than other terms) at the lower levels and large at upper levels. So, to understand the lower level vorticity dynamics, these two terms can be safely ignored.

The stretching term (Figure 9, third from left in the upper panel) is high in magnitude in the convective phase at both upper and lower levels, of course with opposite signs. This is intuitive as this term is expected to be strongly influenced by local divergence. At lower levels, stretching is mostly concentrated below 850 hPa (as convergence is strongest near the surface), and is almost in phase with vorticity anomalies with a slight lag. This lag is consistent with the observation that convection peaks behind or northeast of the cyclonic circulation. Further, like the vorticity anomalies, stretching has the same sign up to 300 hPa. Essentially, the stretching term acts to maintain the low level vorticity anomalies after it reach a peak. We observe that this tendency of the stretching term is somewhat negated by the horizontal advection vorticity anomaly (Figure 9, first from the left in the upper panel), i.e., advection helps in suppressing the low-level negative anomalies after it reach a peak. The $\beta$ effect term (Figure 9, second from the left in the upper panel) leads the vorticity anomalies by about a quarter of a cycle, and also shows a similar tilt at upper levels (above 300 hPa). Hence, $\beta$ contributes to the sign change of vorticity anomalies in the next cycle and is the most likely candidate responsible for its propagation. In all, the view presented by Figure 9, which is an Eulerian one, where, by averaging over a specified box, we see the vorticity anomalies as it pass by a fixed observer, and its successive crests and troughs are led by those of the $\beta$ effect term. Further, the vorticity anomalies are fed and damped at lower levels by the stretching and horizontal advection terms, respectively. Whereas, at upper levels, the terms that mostly cancel each other are the vertical advection and titling, with the former and latter responsible for decay and growth, respectively.

To understand the dynamics at the lower levels in more detail and from a spatial perspective, we present the composite mean of various terms averaged over 850 to 600 hPa to represent the lower level circulation (Figures 10-12). Note that, at near-surface levels the residual term can be quite large in reanalysis (Boos et al., 2015). In fact, Boos et al. (2015) found that the influence



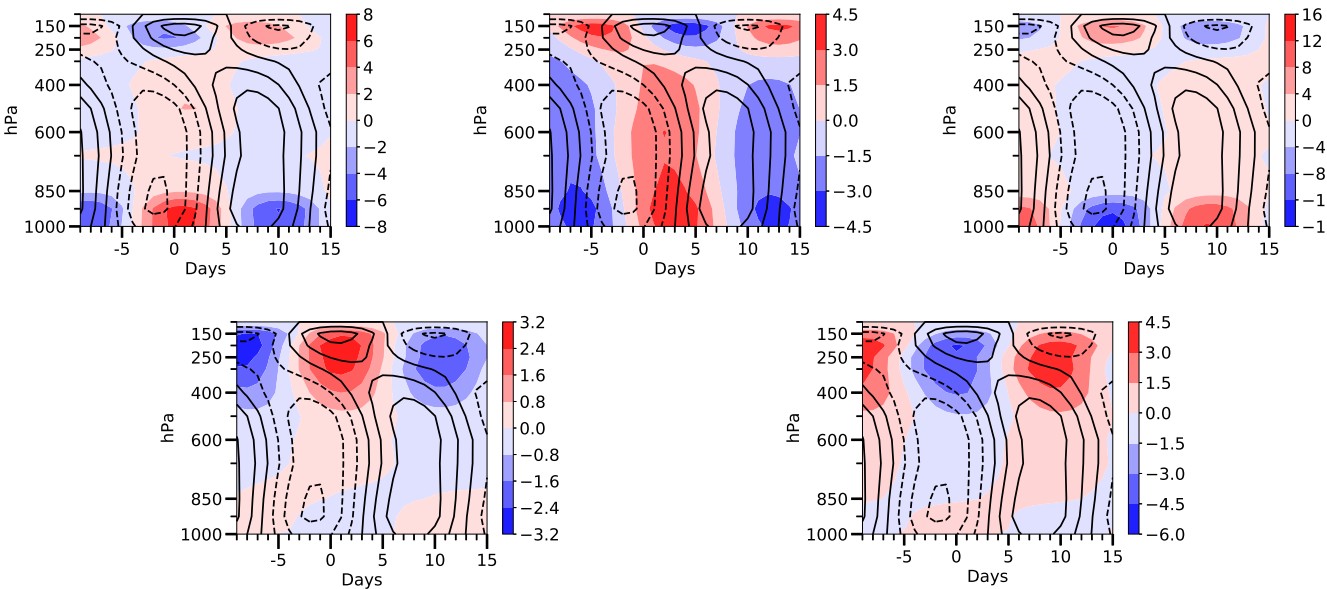

**Figure 9.** Vertical profile of Eulerian time evolution of observed composite mean of 14–30 day anomalies (shaded) with reference to Equation 1(residual term not shown) ,averaged over 10°-17.5°S, 52.5°-60°E (the red box in Figure 1) , shown between 1000–100hPa . Upper panel shows the horizontal advection term, the $\beta$ term and the stretching term (from left), lower panel shows the tilting term and the vertical advection term(from left). Units for all variables are $10^{-11}\mathrm{s}^{-1}$ . The composite mean of relative vorticity anomalies ($\mathrm{s}^{-1}$) averaged over the same box is superimposed for reference (contours). The contour interval for vorticity anomaly is $2 \times 10^{-6}\mathrm{s}^{-1}$, contours signifying positive and zero values are solid, those for negative values are dotted. Note that the terms are plotted with signs as on the RHS of Equation 1.

of this term lessens by about 700 hPa, guided by that observation, we average over 850 hPa to 600 hPa values to represent the lower level circulation. Further, as seen in Figure 9, the major terms of the equation have little vertical variation in these
pressure levels.

In Figure 10, on Day $-3$, we see the Eulerian tendency of relative vorticity of the QBWO has a minimum to the southwest of the cyclonic vortex centre at 850 hPa (marked by a black dot), and maxima in the northeast of the centre. This is consistent with the fact that the cyclonic vortex is moving to the southwest. The only individual term that matches this pattern, that is, has a dipole structure with a minimum in the southwest and maximum in the northeast is the $\beta$ term, though its extent is much larger
spatially than the vorticity tendency. The horizontal advection term is not as strong and has more of a fine scale structure. The stretching term has strong maximum southeast of Madagascar, and a weaker minimum to the northeast. This strong positive stretching term is likely a direct effect of the divergence associated with the anticyclonic circulation that is situated in the same location (Figure 3). The combination of the $\beta$ and the stretching captures the Eulerian vorticity tendency to some extent,





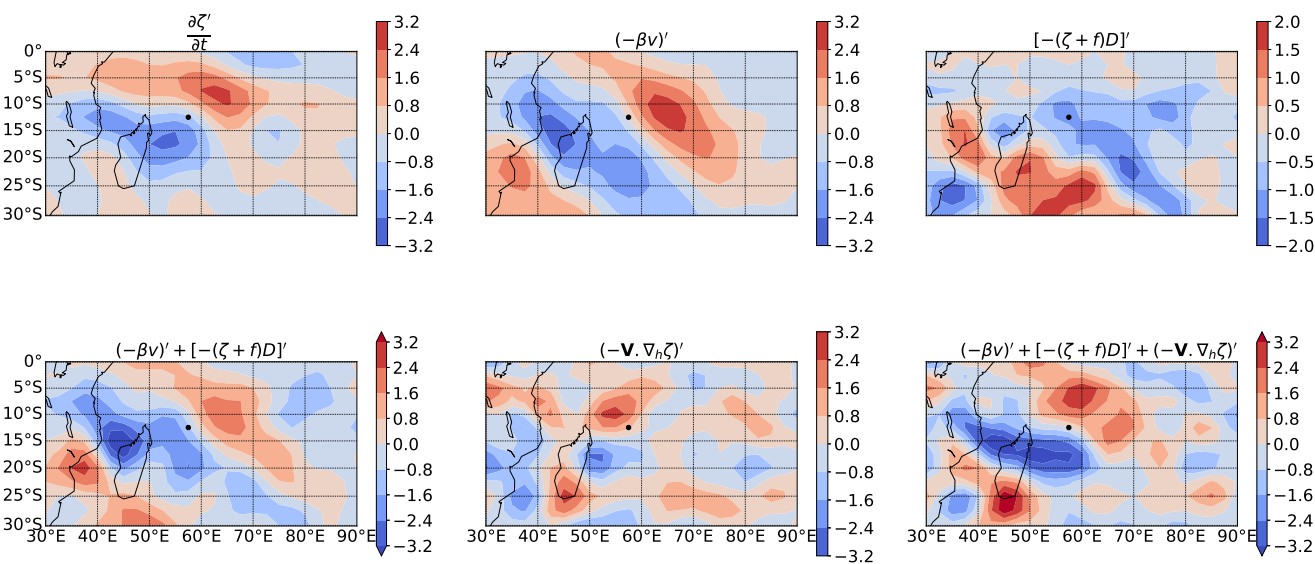

**Figure 10.** Contours of the composite mean of dominant 14–30 day anomaly terms in Equation 1 and their combinations, averaged from 850–600 hPa (lower troposphere) on Day $-3$ of the composite. From left to right in both rows, the terms are: Relative vorticity tendency, $\beta$ term, and Vortex stretching term(Upper row), Stretching term+ $\beta$ term, Horizontal advection term, $\beta$-effect term + vortex stretching term + horizontal advection term(lower row). Units of variables are $10^{-11}\text{s}^{-2}$.The black dot in all the figures marks the position of the cyclonic vorticity centre.

especially to the southwest of the vorticity centre. In fact,the stretching southeast of Madagascar limits the extent of the $\beta$ by
cancelling it out in that region. When we add the horizontal advection term with the aforementioned combination, in spite of some mismatch due to the neglected smaller terms and the residual, the sum is a reasonable approximation to the Eulerian vorticity tendency pattern.

For Day 0 (Figure 11), we see a similar pattern. Now, the vortex centre and associated dipole pattern of the vorticity tendency shift southwest as compared to Day $-3$. Once again, only the $\beta$ term has a similar dipole pattern as the vorticity tendency.
In this occasion, the stretching term is strongly negative to the north and east of the vorticity centre, i.e. , to the eastern side of Madagascar. This is expected as convergence and convection are strongest in that region. This limits the southward extent of the positive $\beta$ term on the eastern side of the centre, and helps the vorticity tendency take the observed shape. Horizontal advection is again comparatively scattered through the domain. The combination of the $\beta$,the stretching and the horizontal advection do not approximate the vorticity tendency as well as on Day $-3$. Though, the rough pattern is captured, with the
NE-SW oriented dipole structure about the vortex centre.





In Figure 12, for Day 3, we observe a similar pattern with the vorticity tendency having shifted further southwest as compared to Day 0. Near the equator, a new negative tendency can be seen northeast of the maxima and is associated with the next low in the QBWO wavetrain. Strong negative stretching is visible east of Madagascar, and it successfully weakens the southern portion of the strong positive $\beta$ term in that region. Here, apart from its southern extent which is a little too large, a combination

of $\beta$ and stretching captures much of the vorticity tendency pattern. Though horizontal advection is scattered over the domain, its inclusion with $\beta$ and stretching helps to capture the overall vorticity tendency pattern.

Overall, from Figures 10-12, we see that the most important factors in defining the shape of vorticity tendency are the $\beta$ and the stretching term. While, at the outset, the $\beta$ term resembles the tendency, stretching limits its southward extent, especially on the eastern side of Madagascar, where the convergence/divergence associated with this QBWO is noted to be strongest (Figure

3). This gives the tendency a more east-west orientation. In other words, the southward movement of this mode would be much faster without strong stretching; so, it can be said that the convectively coupled nature of this mode that gives rise to the strong stretching term essentially slows down the southward movement of the QBWO mode.The horizontal advection, which is not as dominant as these two aforementioned terms mostly plays a damping role and its inclusion leads to a better match with the vorticity tendency.

Figure 13 adopts a Lagrangian perspective and depicts the temporal evolution of composite mean of the dominant terms in the vorticity budget in the lower troposphere (averaged over 850 to 600 hPa) following the track of the system shown in Figure 3. On Day $-9$, a negative vorticity anomaly is visible just south of the equator. Consistent with the Hovmöller diagrams in Figures 4-6, over the course of the next week, the anomaly propagates in a southwest direction. This is followed by a positive vorticity anomaly that appears in a similar location and follows a similar track. As is clear in Figure 13, horizontal advection

is not particularly strong and is scattered inside the vorticity anomaly contours. Though the stretching term is somewhat large, particularly slightly away from the equator, as expected from our discussion of Figure 10-12, it is more or less in phase with the vorticity anomaly, and so it can't be responsible for the propagation of the vorticity anomaly (Wang and Chen, 2017), rather it mainly helps in local enhancement of the vorticity anomaly. The only term that is coherent and leads the vorticity anomaly is $\beta$; in fact, this leads the vorticity anomaly by approximately 4–5 days or about a quarter of a cycle. Interestingly , though there is

a clear lead-lag relation between the $\beta$ term and vorticity anomalies throughout the track, the tendency of $\beta$ term suggests that it is slightly faster than the movement of vorticity anomalies along the track, especially away from the equator. This matches with our observations in Figures 10-12, discussed above. Thus, overall, this lower level along-track perspective is consistent with the vertical structure of the system in Figure 9, and reinforces the role of $\beta$ in the propagation of the QBWO. Once again, these results are consistent with findings regarding the QBWO over the SCS in the boreal summer (Wang and Chen, 2017), but

contradict the claim made by Fukutomi and Yasunari (2013) who probed intraseasonal disturbances moving over the SWIO. They attributed southward movement mainly to the horizontal advection of transient eddy vorticity by the northerly mean flow, but here, in our analysis, the role of horizontal eddy advection is not prominent and the northerly mean flow is absent in the region where the southward movement of the QBWO is most apparent.



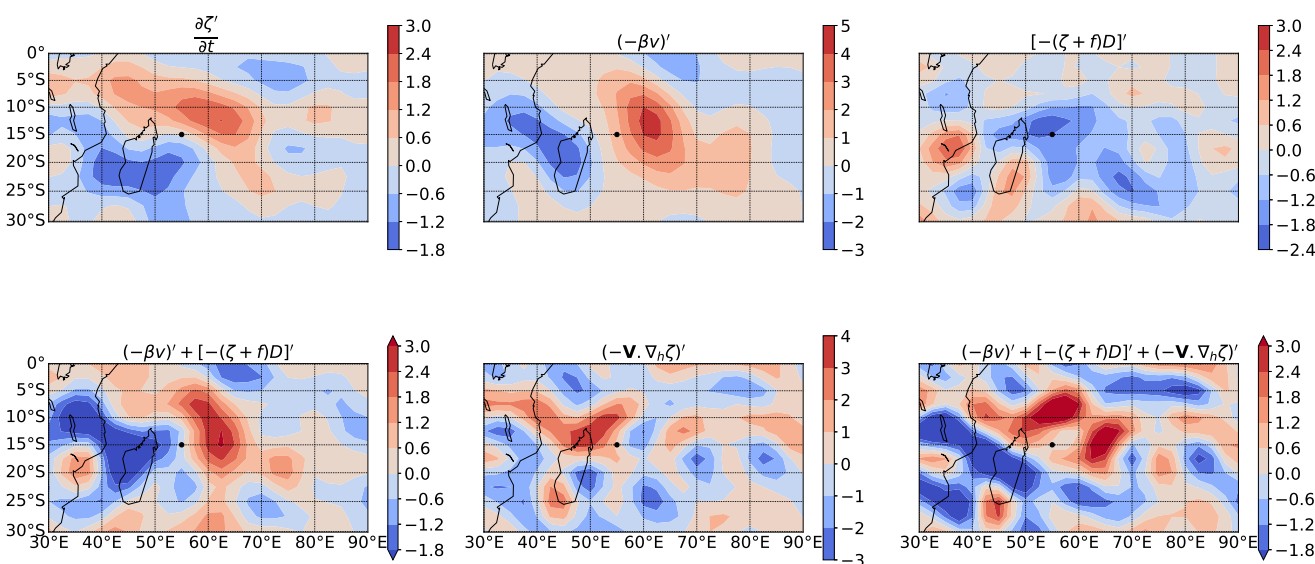

**Figure 11.** Same as Figure 10, but for Day 0.

## 5   A QBWO case study and cyclogenesis

We now explore a particular QBWO episode from the boreal winter season of 2008–09; the aim is to elucidate the propagation and evolution of this system and its connection with the genesis and track of TCs. As per convention, the location of cyclogenesis is defined by the coordinates where a named TC is estimated to attain tropical depression status (Bessafi and Wheeler, 2006). In the following, we show the position of 3 TCs from its depression stage to the point up to which the system retains at least Tropical Storm status as defined by JTWC. Note that we have shown the location of TCs at 0000UTC for a given

day. Till now we have focused on composites of the QBWO and it is instructive to examine particular cases for a look at the details of the systems' formation and propagation. In this context, Figure 14 shows the evolution of the QBWO wavepacket in which consecutive high and low vortices are generated near the equator, and then continue their journey to the southwest in the Southern Hemisphere. The features in Figure 14 are in excellent agreement with those identified in the composite of the QBWO (Figure 3). Note that this particular sequence of disturbances lasted approximately 2.5 months and Figure 14 shows a

specific period from late January to early February of 2009, covering slightly more than one complete cycle of a QBWO.

Maps in Figure 14 are shown in three days increments from January 18 to February 11, 2009. On January 18, we see a northeast-southwest oriented wave of alternating low (around 20°S,50°E; characterized by cyclonic circulation) and high (around 10°S and 70°E; characterized by anticyclonic circulation) pressure signals. The low-pressure region is associated with a large area



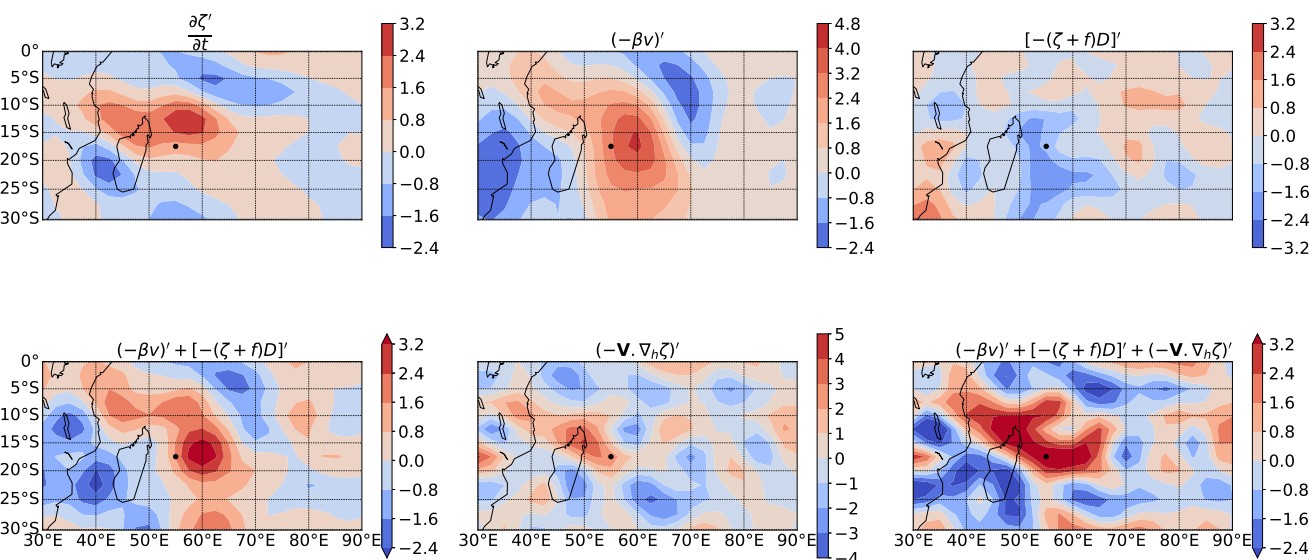

**Figure 12.** Same as Figure 10, but for Day 3.

of above-average convection that covered Madagascar, and below-average convection is associated with the high-pressure area

to the northeast. Inside the low-pressure zone, two nascent tropical depressions can be observed, marked by yellow and orange dots, which eventually became tropical storm Fanele (yellow dot) and Eric (orange dot), respectively. The red contours in this plot are the zone of strongest negative (cyclonic in Southern Hemisphere) vorticity, and clearly, both these storms are within regions of largest vorticity anomalies associated with the QBWO. By January 21, the cyclonic circulation near Madagascar weakened, the anticyclonic vortex with positive OLR anomalies is displaced to the southwest, and to the north we see hints of a

new QBWO low forming just south of the equator (between 60°–90°E), with weak convection and anomalous westerly winds. In this period, Fanele moved very slowly towards the east, Eric moved southward and still remained within the zone of QBWO convection. After 3 days, by January 24, the old cyclonic circulation and the TCs within it disappeared, the anticyclonic lobe moved west-southwestward towards northern Madagascar, and the evidence of a new low with strengthened and expanding negative OLR anomalies and strengthened westerly wind anomalies can be observed just south of the equator.The anomalous

westerly wind also extends above the equator, and one can see a hint of anomalous convection associated with it, albeit weaker than that of the south of the equator. By January 27, the high intensified and moved further southwest engulfing almost all of Madagascar with a large and well-formed anticyclonic vortex and associated positive OLR anomalies. Simultaneously, the newly formed low that emerged near the equator continued to intensify as it moves westward. It is worth noting that the linear shallow water solution for ER waves shows convergence one quarter wavelength east of the low (Matsuno, 1966), this pattern



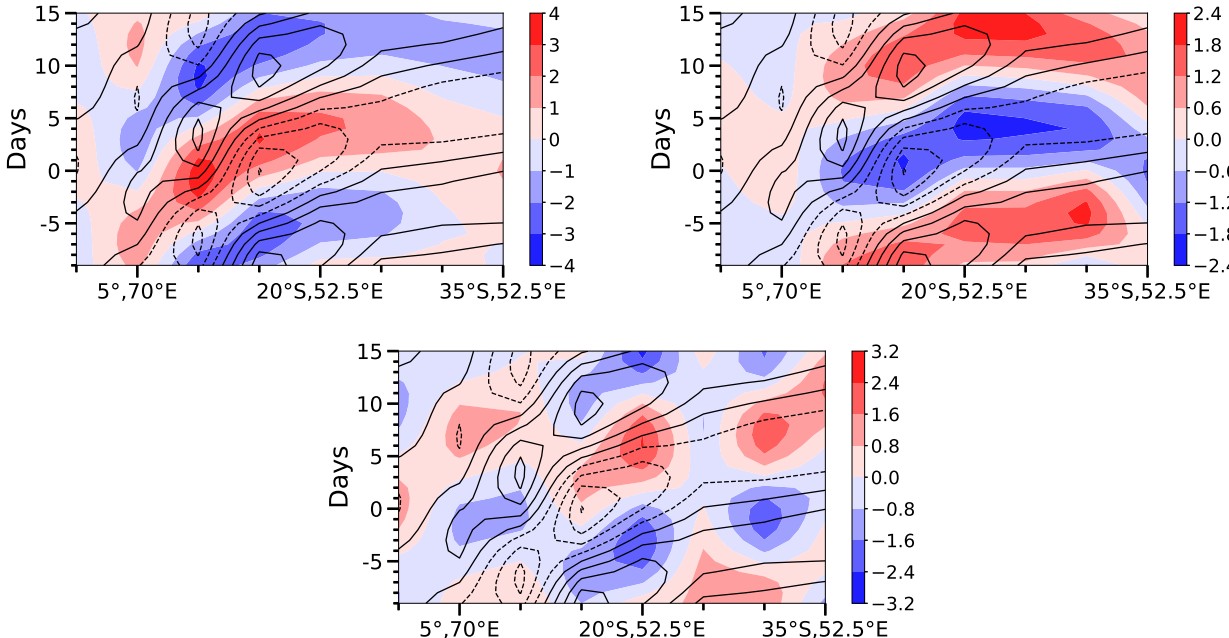

**Figure 13.** Time-evolution of 850 to 600 hPa averaged composite mean of dominant 14–30 day anomaly terms in Equation 1 along the approximate track-line of the system shown in Figure 3.The $\beta$ term (upper left),the Stretching term(upper right) ,the Horizontal advection term(lower). Units for all terms are $10^{-11}\mathrm{s}^{-2}$. The composite mean of relative vorticity anomalies (s$^{-1}$), averaged over 850 - 600 hPa is superimposed for reference (contours). The contour interval for vorticity anomaly is $2 \times 10^{-6}\mathrm{s}^{-1}$, contours signifying positive and zero values are solid, contours signifying negative values are dotted.

is reflected to some extent with strong negative OLR anomalies east of the low; a feature that has also been observed in ER waves (Molinari et al., 2007) and the boreal summer QBWO (Chen and Sui, 2010). The above-average convection over the equator is still visible, though it remained almost stationary, and then decayed in the next few days.

The dominant pattern in the south of the equator described above repeats through the following days shown in Figure 14, though of course with the sense of anomalies reversed in the next cycle of the oscillation. More strikingly, on February 2, another

tropical depression (eventually TC Gael) can be seen inside the low near 15°S, 70°E, i.e., within the strong negative OLR anomalies in the northeast sector of the low-pressure lobe of the QBWO. Like Fanele and Eric, this system also formed within the region of strongest vorticity anomalies. By February 5, the low intensified further, expanded in scale with a well-defined cyclonic vortex, moved further southwestward and convection associated with it covered most parts of Madagascar and made inroads into southeast Africa. Concomitantly, the depression intensified into a TC (named Gael) and moved southwestward.

As the QBWO wavepacket evolved, TC Gael moved southwards inside its circulation gyre towards the coast of Madagascar (February 8). Finally, by February 11, i.e., in the last subplot of Figure 14, the cyclonic anomaly near Madagascar almost





disappeared and TC Gael moved towards the southeast and transitioned to an extratropical disturbance (not shown). In greater detail, Figure 15 shows unfiltered OLR and winds at 850 hPa at 0000 UTC of February 5 and February 8, and contours of 14–30 days filtered OLR anomalies to showcase the movement of TC Gael under the influence of QBWO. In fact, on February 5, almost the entire TC with its strongest winds can be seen within the outer contour of 14–30 days filtered OLR, and the core is inside the inner contour of the negative QBWO lobe. On February 8, negative OLR anomalies associated with this low are weaker, still, most of the TC is contained inside the outer contour of these anomalies, with the centre still inside the inner contour.

In addition to these three TCs, as seen in Figure 16, other depressions that matured into TCs were also formed within this QBWO wavetrain. Specifically, TCs Cinda, Dongo, Hina and Izilda can be seen in the negative OLR lobes of the QBWO wavetrain on December 15, January 9, February 21 and March 24 respectively[1] (marked with red, orange, yellow and gold color dots, respectively, in Figure 16). These four cyclones were born as depressions inside large anomalous vorticity zones associated with the QBWO lows; this is made clear in Figure 16 where regions of strongest cyclonic vorticity are marked with red contours. Together, all seven depressions considered here originated inside negative OLR anomalies of less than $-10 \text{W/m}^2$, and in regions of strong vorticity, thus satisfying most of the criteria used to assess whether a storm is associated with ER activity on a quasi-biweekly timescale (Molinari et al., 2007) [2]. Further, we also note that all seven cyclogenesis events described here happened inside the region of large background cyclonic vorticity (Figure 7). Even though this examination was of a particular winter season where QBWO activity was exceptionally long-lasting, we have observed the birth of depressions and their maturity into TCs inside the QBWO wavetrain in numerous years. For example, one of the most intense TCs in the recorded history in this basin, Cyclone Gafilo in 2004, was also born and moved under the influence of a QBWO wavetrain. As seen in the composite in Figure 3 and 7, convectively coupled QBWO lows become strongest in regions of high background vorticity, thus superposition of the background cyclonic circulation and QBWO cyclonic circulation possibly makes the area very conducive for cyclogenesis. It is worth mentioning that there are also cases where depressions that matured into cyclones were born outside the QBWO envelope, but eventually their movement was influenced by the environmental quasi-biweekly circulation anomalies.

More broadly, as is known, low-level vorticity and vertical wind shear can be modulated by intraseasonal oscillations, especially equatorial waves, thus, these oscillations can affect subseasonal variability of cyclogenesis (Maloney and Hartmann, 2000; Frank and Roundy, 2006; Bessafi and Wheeler, 2006; Molinari et al., 2007; Camargo et al., 2009; Schreck et al., 2012). Additionally, OLR anomalies associated with waves serve as pre-existing convective disturbances from which TCs can be born, and the mean upward motion associated with moist convection can further aid cyclogenesis (Frank and Roundy, 2006; Bessafi and Wheeler, 2006). In the SWIO, the influence of gravest ER wave modes on cyclonic activity has been noted, and the modulation of cyclogenesis is most readily attributable to large variations in low-level, off-equatorial vorticity and OLR by these ER waves (Bessafi and Wheeler, 2006). The previous sections emphasized similarities between the QBWO

---

[1]Here, we have shown the plots of the days when the tropical depressions are first detected at 0000UTC.

[2]If a TC is not born by 0000UTC of the days shown here, rather it starts as a depression at an earlier time, we confirm it's genesis location still satisfies the criteria mentioned above. This also applies for the TCs shown in the case study plot of Figure 14.




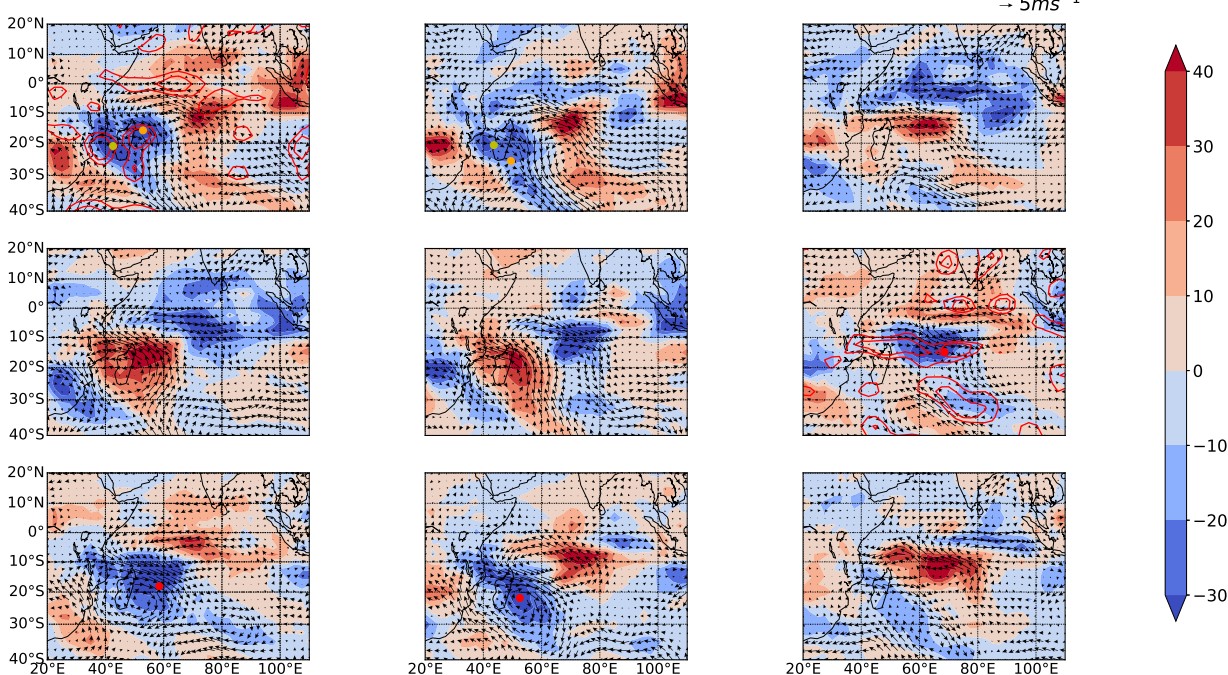

**Figure 14.** 14–30 day filtered OLR (W m$^{-2}$) and 850 hPa wind anomalies from 18 th January to 11th February(with an interval of 3 days) of the boreal winter season (DJFM) of 2008-09 . Specifically, January 18, 21, 24 (first row, from left); 27, 30, February 2 (second row, from left) and 5, 8 and 11 February (third row,from left) of 2009 are shown. Yellow, orange and red dots in individual subplots (Jan 18, 21 and Feb 2, 5, 8) are the locations of TCs Fanele, Eric and Gael at 0000UTC of the respective days, respectively, the dots are shown from tropical depression stage onward, as defined by the Joint Typhoon Warning Center (JTWC). Red contours are of 14–30 day vorticity on days when the tropical depression can be first detected at 0000UTC. Only negative (cyclonic in Southern Hemisphere) vorticity is contoured, the value of the outer contour is $-4 \times 10^{-6}$s$^{-1}$, and the inner contour is $-8 \times 10^{-6}$s$^{-1}$.

and the gravest ER wave which is known to influence cyclogenesis in SWIO (Bessafi and Wheeler, 2006). Indeed, much

like the sequence of events depicted for the seven TCs within the QBWO in 2008–09, Bessafi and Wheeler (2006) found that cyclogenesis that happens during off-equatorial ER wave activity occurs inside the cyclonic vorticity and convective anomalies. Further, they also noted that, once born, systems move systematically westward with the ER wave (indeed, TCs Gael, Eric, Cinda and Hina initially moved westward). Thus, much like the connection of QBWO to cyclogenesis in other tropical basins (Ling et al., 2016; Zhao et al., 2016), here too, quasi-biweekly variability clearly influences the birth and evolution of TCs.


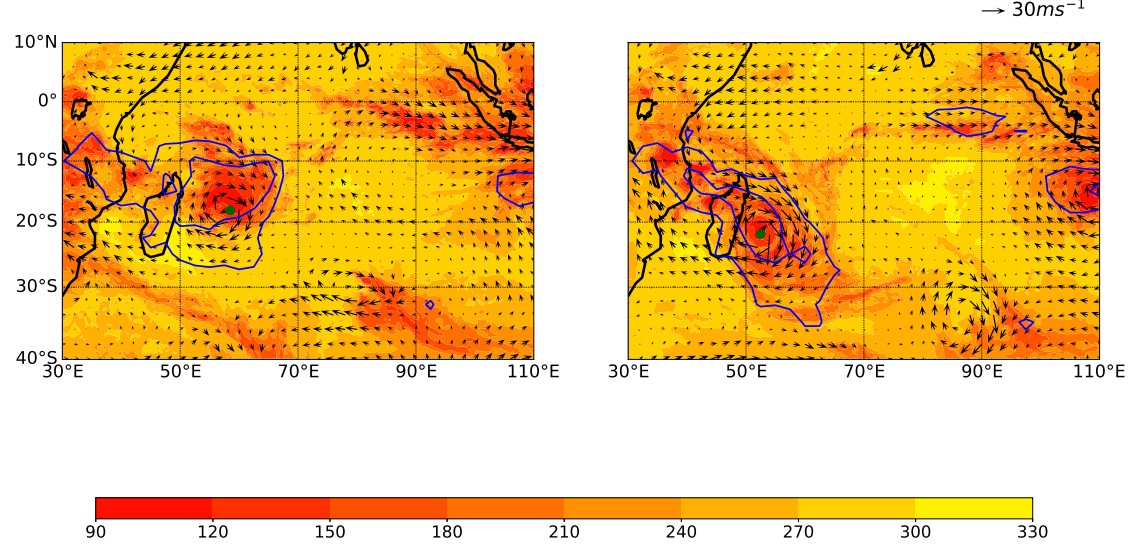

**Figure 15.** Unfiltered OLR (W m$^{-2}$) and 850 hPa wind at 0000 UTC on February 5 and February 8 of the boreal winter season (DJFM) of 2008-09. Blue contours indicate 14–30 day filtered OLR anomalies of the respective days. Only negative (cyclonic in Southern Hemisphere) anomalies are contoured, the value of the outer contour is $-15$ W m$^{-2}$ and the inner contour is $-30$W m$^{-2}$. The green dot in both the figures are the location of Cyclone Gael at respective days.

## 6 Conclusions

In this study, we have identified a convectively coupled quasi-biweekly signal in the SWIO during the boreal winter. At the outset, we note that regular quasi-biweekly oscillations are observed almost every year in this region, usually during the second half of the winter season. In fact, this intraseasonal variability in the SWIO is contained within an envelope of high precipitable water. A composite drawn from 10 years of data shows that the QBWO exhibits distinct propagation characteristics in filtered OLR and low-level circulation. Specifically, the system starts near the equator, west of Sumatra, and then propagates southwest towards Madagascar. After crossing Madagascar, the QBWO weakens and eventually disappears near $30°$–$35°$ S. On average, the QBWO gyres have a northwest-southeast tilt, a westward phase speed of about 4 m s$^{-1}$, the system has a time-period of about 18-20 days and a zonal planetary wavenumber 5-6 (negative OLR anomaly lobes as well as associated circulation gyres span approximately 30–35 degrees in longitude and 20 degrees in latitude). A pronounced feature of the QBWO is that the strongest moist convection occurs in the northeast sector of its cyclonic circulation and is probably due to both convergence and poleward rotational advection of moist air.



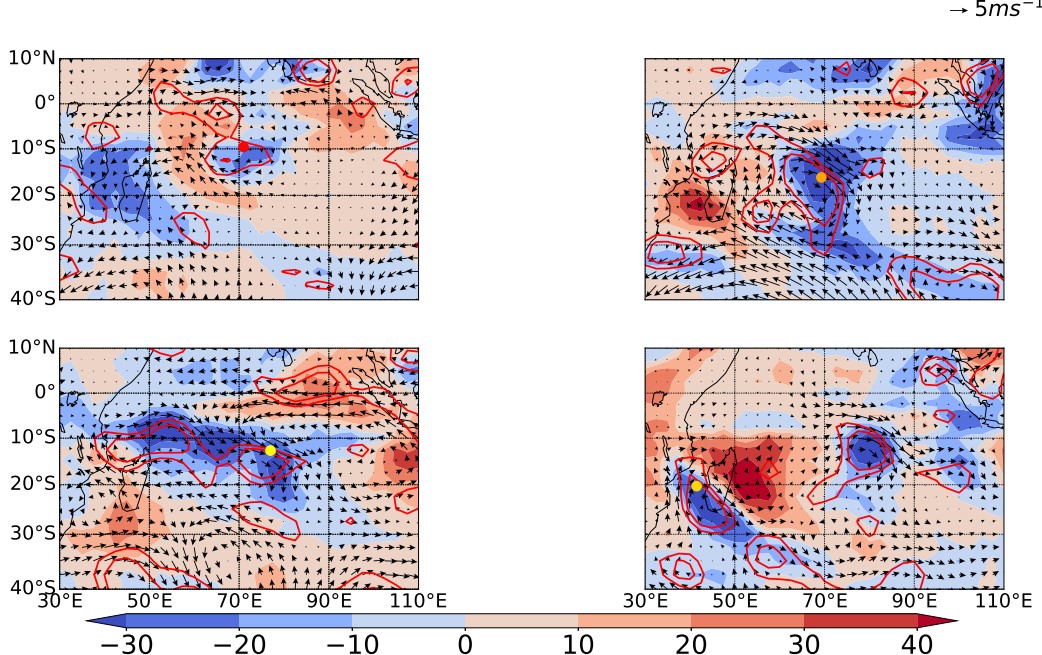

**Figure 16.** 14–30 day filtered OLR (W m$^{-2}$) and 850 hPa wind anomalies for December 15, January 9 (upper left and right respectively), February 21 and March 24 (lower left and right respectively) of boreal winter season of 2008-09. Red, Orange, Yellow and Gold dots in individual subplots are the locations of the centre of TCs Cinda, Dongo, Hina, and Izilda, respectively at 0000UTC for the respective days. The plots are shown for the day when the tropical depression was first detected at 0000UTC, as defined by the JTWC. Red contours are of 14–30 day vorticity anomalies of the respective days. Only negative (cyclonic in the Southern Hemisphere) anomalies are contoured, the value of the outer contour is $-4 \times 10^{-6}\mathrm{s}^{-1}$, and the inner contour is $-8 \times 10^{-6}\mathrm{s}^{-1}$.

The vertical structure of the QBWO shows a nearly barotropic structure when it is initially observed near the equator, a first baroclinic mode profile emerges as the system couples with moist convection and slowly moves away from the equator, and finally, by about 35° S, an equivalent barotropic structure re-emerges when the oscillation weakens at the end of its life-cycle.

Further, near the equator, vertical velocity and vorticity extrema are not aligned. In fact, the maximum (minimum) vertical pressure velocity is located to the northeast of vorticity maximum (minimum), whereas away from the equator, these two are aligned. With regard to the dynamics at play, a vorticity budget analysis showed that the $\beta$ effect is the main contributor in the southwestward propagation of QBWO vorticity anomalies. The roles of stretching and horizontal advection are also important to determine the spatial structure of the vorticity anomalies in the lower troposphere. Further, stretching, due to

convergence/divergence slows down the southward propagation speed of the QBWO. Thus, we see the physical manifestation of moist convective coupling in reducing the speed of this mode. Further, this analysis allows us to hypothesize that differential planetary rotation, coupling with moisture and horizontal rotational advection are the minimal ingredients required for the QBWO.





These features, i.e., the wavenumber, phase speed, and time-period fall within the range reported for convectively coupled

gravest ER waves (Wheeler and Kiladis, 1999; Wheeler et al., 2000; Kiladis et al., 2009; Janicot et al., 2010). Moreover, the position of strongest convection is also similar to observations of ER waves (Wheeler et al., 2000; Molinari et al., 2007), and agrees with simplified moist shallow water simulations (Suhas and Sukhatme, 2020). The baroclinic nature of the QBWO while strongly coupled with convection, and near-barotropic structure while loosely coupled/uncoupled with convection are also observed in ER waves (Wheeler et al., 2000; Yang et al., 2007; Kiladis et al., 2009). While this agreement with ER

waves is encouraging, certain features need to be kept in mind. Quite conspicuously, the QBWO is restricted to the Southern Hemisphere in this season. Though idealized ER waves are symmetric about the equator (Matsuno, 1966), and studies have observed symmetric ER waves (Kiladis and Wheeler, 1995; Chatterjee and Goswami, 2004), this kind of asymmetric wave structure which is purely dominated by one hemispheric component is anticipated and documented in different background flows (Wheeler et al., 2000; Molinari et al., 2007; Chen and Sui, 2010). With regard to propagation, the QBWO moves westward

and poleward, whereas ER waves on a background state of no flow are equatorially trapped. In reality, Rossby waves triggered in the tropics can leave the equatorial wave-guide under the influence of non-zero zonal mean flows (Lau and Lim, 1984; Sardeshmukh and Hoskins, 1988; Zhang and Webster, 1989; Jin and Hoskins, 1995; Molinari et al., 2007; Monteiro et al., 2014). In fact, this poleward motion is consistent with the propagation of the boreal summer QBWO (Chatterjee and Goswami, 2004; Kikuchi and Wang, 2009; Chen and Sui, 2010; Wang and Chen, 2017). Taken together, the structure, features and

dynamics of the QBWO in the boreal winter has many similarities with its Northern Hemisphere counterparts, both of which appear to be consistent with convectively coupled ER waves that are modified by suitable background states.

Though the principal features of the QBWO match with convectively coupled ER waves, the genesis of these oscillations requires further study. Convective feedback mediated by boundary layer convergence does take a step in this direction (Chatterjee and Goswami, 2004), and idealized models also suggest that easterly vertical shear traps the ER wave in the lower troposphere

and helps growth through diabatic heating (Wang and Xie, 1996; Xie and Wang, 1996). Further, when vertical shear is asymmetric about the equator, the unstable ER wave is constrained to the hemisphere where the easterly shear is prominent (Wang and Xie, 1996; Xie and Wang, 1996). Such a vertical shear is present between $0°$-$10°$S in the SWIO where the QBWO starts to grow. In addition to the winds and shear, the concentration of PW in the Southern Hemisphere in the boreal winter possibly helps to facilitate the coupling with moisture south of the equator in this season. Indeed, building on the interaction of moisture

gradients with waves (Sobel et al., 2001; Sukhatme, 2014; Monteiro and Sukhatme, 2016), recent work has uncovered newer routes of genesis such as moisture-vortex instability (Adames and Ming, 2018; Adames, 2021) or enhancement of barotropic processes (Diaz and Boos, 2019, 2021) that can play a role in the growth of large-scale moist systems. Thus, a holistic theory for the QBWO, will, in addition to the minimal ingredients listed above, possibly include interaction with seasonal background shear and moisture gradients. It is worth noting that general circulation models still have difficulties in producing an accurate

QBWO (Jia and Yang, 2013; Wang and Zhang, 2019). An understanding of the basic mechanisms involved in the QBWO will help diagnose the issues that plague these efforts, and will contribute to enhanced intraseasonal prediction capabilities, especially in the context of Madagascar and parts of south-east Africa, where the boreal QBWO has significant influence.



Finally, convergence, moist coupling and large vorticity anomalies of the QBWO generate a favourable environment for the birth of tropical depressions, some of which go on to intensify in cyclones. Specifically, using the case study of 2008–09 winter,

we showed the repeated genesis of depressions inside QBWO lows, preferentially in the NE sector.We have shown seven of these systems that evolved into TCs. In particular, the influence of the QBWO environment on the birth and propagation of TC Fanele, Eric and Gael were examined in detail. Indeed, the convective environment provided by the QBWO in conjunction with the vorticity of the background flow appears to provide the most suitable conditions for cyclogenesis. The connection between ER waves and TC genesis is well established (Frank and Roundy, 2006; Bessafi and Wheeler, 2006; Molinari et al.,

2007; Schreck et al., 2012), and the interpretation of the QBWO as a suitably modified grave ER mode adds merit to the possibility of cyclones preferentially forming within the zone of strong cyclonic vorticity anomaly and convection associated with the Rossby gyres. In the SWIO, Bessafi and Wheeler (2006) have reported instances of cyclogenesis in both east and west of the ER low, but the most probable region is again northeastward of the centre of the ER low which is in agreement with our observation of cyclogenesis in the QBWO. Though we focused on one season, it is important to note that the genesis of

depressions and their conversion to TCs within the QBWO envelope was observed in other seasons too. In fact, the symmetry imposed on ER wave detection (Bessafi and Wheeler, 2006) can be relaxed due to the hemispheric preference of the QBWO, and this can potentially lead to the identification of more systems spawned in these moist convective envelopes, thus enhancing the predictability of a larger number of depressions and cyclones in the SWIO.

*Code and data availability.* All data used in this study is publicly available. NCEP-NCAR reanalysis can be accessed from https://psl.noaa.

gov/data/gridded/data.ncep.reanalysis.html, NOAA OLR data can be accessed from https://psl.noaa.gov/data/gridded/data.interp_OLR.html, ERA5 data can be accessed via https://www.ecmwf.int/en/forecasts/datasets/reanalysis-datasets/era5, ERA-Interim datas can be accessed via https://www.ecmwf.int/en/forecasts/datasets/reanalysis-datasets/era-interim, TRMM data can be accessed via https://disc.gsfc.nasa.gov/datasets/TRMM_3B42_7/summary, JTWC best track data can be accessed via https://www.metoc.navy.mil/jtwc/jtwc.html?best-tracks, and Kalpana satellite OLR data can obtained from https://tropmet.res.in/~mahakur/Public_Data/index.php?dir=K1OLR. Details for accessing

and using the Windspharm package are given at https://ajdawson.github.io/windspharm/latest/.

*Author contributions.* Both authors contributed equally to the work.

*Competing interests.* The authors declare no competing interests.

*Acknowledgements.* SG acknowledges financial assistance from Council for Scientific and Industrial Research (CSIR) and the Divecha Centre for Climate Change, IISc. JS acknowledges support from the University Grants Commission, project F 6-3/2018 under the Indo-Israel

Joint Research Program ($4^{th}$ cycle).





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
