# Peer review of "Southwestward propagating quasi-biweekly oscillations over the South-West Indian Ocean during boreal winter"

_Weather and Climate Dynamics, 2021_

## Author Comment (AC1)

**Response to Reviewer 1**

(Please note that the Reviewer's comments are in normal font and our response is in italics)

The authors present evidence for the existence of a quasi-biweekly oscillation in the southern Indian Ocean. They show some composites, a vorticity budget analysis, and show how the mode may modulate TC activity. If confirmed, this study could reveal our understanding of these quasi-biweekly modes. However, I have multiple concerns about the contents of the manuscript, which I outline below. Because of this I recommend major revisions.

*We thank the reviewer for encouraging comments on our work. We have addressed the concerns of the reviewer below.*

Main comments:

1. The manuscript feels long and disjointed. There is a lack of organization of the manuscript which makes reading it difficult and exhausting. Mean state plots should be shown together, as well as the plots about the vorticity budget. Some plots could be coalesced to save space or even gain new insights. There is also some parts of the manuscript that feel unnecessary or are not well-justified. For example, why discuss equatorial Rossby waves in the Introduction? The mode shown here is not an equatorial Rossby wave. If there's a point to this, the authors should be more clear about it. Overall, I think the authors can trim a lot of the content that is currently shown and focus on the essentials, as well as focus more on the two other major concerns below.

*We thank the reviewer for this criticism. We have now trimmed the manuscript in various places and we hope this has improved the flow of the narrative. The mean flow and vorticity have now been moved to Figure 2 (i.e., following the mean PW and OLR variability in Figure 1). In all, the literature on intraseasonal variability in this region is quite sparse, thus, while preparing the manuscript we felt it would be better to include a broader background rather than only focusing on our findings. We appreciate that we probably went overboard with this, and as mentioned, have streamlined the presentation of the material in the revised manuscript. Further, the equatorial Rossby wave — specifically, the moist equatorial Rossby wave — provides an underpinning for understanding the QBWO. Indeed, past literature has interpreted the QBWO as a modification of the gravest equatorial Rossby wave. Hence, we choose to highlight similarities in the two modes whenever possible.*

2. Statistical significance of the mode: The analysis shown here is based on a composite analysis on a box over the southwest Indian Ocean. It is unclear why this box was chosen, and no attempt is made to show that the quasi-weekly mode is statistically significant. This could be done by showing that the power spectrum of vorticity or OLR is above the red spectrum at the 99% confidence interval. An EOF analysis showing that the eigenvalues corresponding to this mode are statistically distinct could also be shown. However, the analysis as shown in its current form is not sufficiently convincing. This is important, as it is otherwise unclear why the authors chose the filtering process outlined in the paper – it seem ad hoc.

*Statistical significance of the composite is now included in the revised presentation. The QBWO mode shown in the revised figures is statistically significant at 95% confidence level. The results have not changed much when we have plotted only statistically significant signal in the revised manuscript. Indeed, the signals are much clearer and we thank the reviewer for this suggestion. The comparison to red noise has already been performed by Kikuchi & Wang 2009. We should have referenced this in the text and have now done so. Indeed, a signal above red noise in the 12–20 day band (in south-east Africa and the southern equatorial Indian Ocean), that accounts for about 8% of the variance in a empirical orthogonal mode decomposition*

*of OLR anomalies in the SWIO was demonstrated in Kikuchi & Wang 2009. As already mentioned, the box chosen has been varied to check for the robustness of results. Indeed, altering it within the pocket of high OLR variability in the southwest Indian Ocean does not affect the results.*

3. The vorticity budget is not enough to justify the main points of the paper. There is discussion about moisture advection throughout the paper yet not discussion about a moisture budget. This should be included. Even better would be an MSE budget or a weak-temperature gradient balance-based moisture budget (see Chikira 2014, Wolding et al. 2016, Adames and Ming 2018a). The authors should also check whether the water vapor explains most of the precipitation variance. On when examining the evolution of moisture can we better understand how convection is modifying the evolution of the vortex.

*Thank you. We have seen that in this mode, convective variability moves in unison with column-integrated specific humidity (Figure 14 of the revised manuscript). So, a moisture budget analysis is appropriate to understand the evolution of the convective signal. The moisture budget refines our understanding of the QBWO, and as the reviewer can probably guess, a combination of the vorticity and moisture perspectives is what is needed to truly capture a mode like the QBWO. With regard to moisture, its tendency is not accounted by any individual term. Within the QBWO gyres, vertical advection balances the apparent moisture sink. The net moistening due to the combination of these two terms moistens the region in front of the convection, and qualitatively captures the tendency. Though, horizontal advection is not prominent, it leads to moistening over the Mozambique channel and adjacent coast of Africa. In particular, advection of the background moisture by the eddy component of the horizontal flow is observed to be important in the growth of moisture anomalies. Our, longer term goal in this project is to use these observational results to build on a moisture mode like paradigm for the QBWO. The individual vorticity and moist parts will form the basis for such an exploration.*

Minor Comments:

1. Figures: The contents of the figure should be shown in the title. The color bars should say what fields its showing, and the abscissa and ordinate should be labeled. They are not labeled in most figures.

*On the suggestion of the reviewer we have brought in titles for Figures 4, 10, 11 & 12. We have also have also included the content names for composite figures and figures where many terms and their combinations are shown, and following their sequence is necessary. The axes are now labelled (except where it is latitude and longitude, which are obvious from their corresponding unit) in each figure, though not in each subplot as that seemed repetitive.*

Figure 1: Grid lines are obstructive. Consider removing.

*We have made these lighter in color and more sparse. The presence of these lines is useful in our opinion for providing a ready scale of reference.*

Fig. 3: Same comment as Fig. 1. Lags should be shown in the title to make it easier and more intuitive for the reader.

*Figure 3 is now Figure 4 in the revised manuscript. See response above to minor comment 1.*

176-177: outlined in the Data and Methods section. This part of the sentence is unnecessary. Please remove.

*Removed.*

---

## Author Comment (AC2)

**Response to Reviewer 2**

(Please note that the Reviewer's comments are in normal font and our response is in italics)

This paper documents the convectively coupled quasi-biweekly oscillation (QBWO) in the South west Indian ocean. The paper is easy to read and is rather descriptive in nature. The introduction is expansive and provides relevant background on the topic and ends with a clear statement of the goals of the paper. The discussion section recaps some of the physical mechanisms of the genesis of this oscillation.

Despite some promising initial discussion of background moisture distribution (the authors appeared to hint at some moisture mode type behavior) a vorticity budget was the route taken here. This does not address the organization and modulation of convection (perhaps a moisture or moist static energy budget would be useful for that). The key result here is that planetary vorticity advection accounts for the propagation of the wave and stretching to its amplification. The former is consistent with the notion that the wave is an ER type mode and the latter points to the vorticity generation by convergence/divergence associated with convection.

In section 5, the paper presents some material on tropical cyclone formation during the QBWO of 2008–2009. This is also easy to read and is again descriptive in its treatment with no calculations or diagnostics (beyond maps of filtered fields)

Over all, the paper provides documentation of the QBWO in a basin that has not received as much attention as compared to other basins. The results are not necessarily novel but will be useful reference points for future work (such as evaluation of theoretical and conceptual models of this phenomenon).

*We thank the reviewer for the constructive comments. Indeed, our longer term goal in this project is to build towards a moisture mode like paradigm for the QBWO. The present work lays the foundation for this. As the reviewer has suggested, we have now included a moisture budget which refines our understanding of the mode along with our vorticity budget analysis.*

Other comments:

1. The authors might wish to consider calculating statistical significance for their composite anomalies and only show values that are deemed significantly different from zero.

*Statistical significance of the composite is included in the revised presentation. In fact, the composite figures have been revised and all values presented are statistically significant at a 95% confidence level. The comparison to red noise has already been performed by Kikuchi & Wang 2009. We should have referenced this in the text and have now done so. Indeed, a signal above red noise in the 12–20 day band (in south-east Africa and the southern equatorial Indian Ocean), that accounts for about 8% of the variance in a empirical orthogonal mode decomposition of OLR anomalies in the SWIO was demonstrated in Kikuchi & Wang 2009. As already mentioned, the box chosen has been varied to check for the robustness of results. Indeed, altering it within the pocket of high OLR variability in the southwest Indian Ocean does not affect the results.*

2. The data and methods seem reasonable

*Thank you.*

3. Line 160: Just to be sure, can you add a few lines (connecting constant phase) on Fig. 2 to illustrate the wave (phase) propagation. Can you also estimate the southward phase speed and check if they are realistic and the patterns in the Hovmoller represent propagation.

*The guiding lines for the phase speed are shown in Figures 5,6 & 7. Indeed, these allow for a estimation of the speeds in different directions.*

4. Line 165: Same as above, but for the group speed.

*See response above.*

5. Line 207: How does an oscillation die? Is it being damped or absorbed by the background flow? Or is the "weakening" of the composite anomalies simply because one is averaging a band-passed field many days away from the reference time (lag 0).

*The moist signal fades out as convective coupling in this mode dies out by about 35S. The reviewer's questions is very astute in that a dry signal can continue propagating without the accompanying moist anomaly. We see that the vorticity anomalies tend to merge in with the midlatitudes and be "carried away" from this region. We have not studied the implications of this tropical midlatitude interaction, though it would be a very interesting aspect to pursue.*

6. Figure 8: Any idea why the structure changes from 1st baroclinic to a tilt? Is it really tilting or is that simply an artifact of the contouring/shading?

*The first baroclinic signal fades as convective coupling weakens. In addition, the vertical shear in the background flow plays a role in the observed tilting of the system.*

7. Line 484: OLR anomalies are the visible outcome of moist convection. Please rephrase this sentence to make it less redundant.

*Removed.*

8. Section 5 on the impact of the QBWO on troical cyclone formation is again very qualitative in the way it is presented. No real issues here but a more comprehensive study would need simulations with a full physics model and sensitivity experiements.

*We certainly agree with the reviewer here. This is a first cut at the issue that clearly shows a qualitative association between the QBWO and cyclogenesis. A simplified or full model study would be required to probe the dynamical mechanisms involved and the pathway of nurturing cyclones.*